# Expressive Value Learning for Scalable Offline Reinforcement Learning

## Abstract

Reinforcement learning (RL) is a powerful paradigm for learning to make sequences of decisions. However, RL has yet to be fully leveraged in robotics, principally due to its lack of scalability. *Offline* RL offers a promising avenue by training agents on large, diverse datasets, avoiding the costly real-world interactions of *online* RL. Scaling offline RL to increasingly complex datasets requires expressive generative models such as diffusion and flow matching. However, existing methods typically depend on either backpropagation through time (BPTT), which is computationally prohibitive, or policy distillation, which introduces compounding errors and limits scalability to larger base policies. In this paper, we consider the question of how to develop a scalable offline RL approach without relying on distillation or backpropagation through time. We introduce *Expressive Value Learning for Offline Reinforcement Learning* (EVOR): a scalable offline RL approach that integrates *both* expressive policies *and* expressive value functions. EVOR learns an optimal, regularized $Q$-function via flow matching during training. At inference-time, EVOR performs inference-time policy extraction via rejection sampling against the expressive value function, enabling efficient optimization, regularization, and compute-scalable search *without retraining*. Empirically, we show that EVOR outperforms baselines on a diverse set of offline RL tasks, demonstrating the benefit of integrating expressive value learning into offline RL.

## 1 Introduction

Reinforcement learning (RL) is a powerful paradigm for learning to make sequences of decisions, having been successfully applied to the fine-tuning of pretrained large language models (LLMs). However, the success of RL in the language domain has yet to be matched in robotics. In contrast to the language setting, robot interactions occur in the real world, which can be costly, time-consuming, and may pose safety concerns. The constraints of real-world RL naturally motivate the *offline* RL setting, where an agent attempts to learn from a diverse, often sub-optimal dataset *without* further interaction with the environment.

Offline RL scalability spans three primary axes: (1) *scaling data*, (2) *scaling models*, and (3) *scaling compute*. To scale data, offline RL algorithms must learn from larger, more diverse datasets that are often sub-optimal and multi-modal (e.g., generated by multiple policies of varying quality). The need to model such complex data distributions necessitates more powerful models. A promising direction for scaling offline RL is to leverage powerful, expressive generative models such as diffusion (Sohl-Dickstein et al., 2015; Ho et al., 2020; Song et al., 2021) and flow matching (Lipman et al., 2024; Esser et al., 2024).

Existing offline RL methods using generative models typically treat the generative model as the policy, enabling richer action distributions than standard Gaussian-based policies in continuous control (Hansen-Estruch et al., 2023; Chen et al., 2023; Ding & Jin, 2023; Wang et al., 2022; Espinosa-Dice et al., 2025; Park et al., 2025b; Zhang et al., 2025). At a high level, diffusion and flow-based RL policies generate actions through an iterative denoising process, which requires backpropagating through the sampling steps (i.e., backpropagation through time). Backpropagation through time is computationally expensive, memory-intensive, and can degrade the general knowledge of the underlying base policy (e.g., a vision-language model (VLM) in the vision-language-action (VLA) setting) (Ding & Jin, 2023; Zhou et al., 2025b;c).

As an alternative to backpropagation through time, distillation-based methods compress the multi-step policy (e.g., a standard diffusion or flow model) into a one-step model, which can be more efficiently optimized through standard policy gradient techniques (Ding & Jin, 2023; Chen et al., 2023; Park et al., 2025b). However, distillation-based

methods face a fundamental limitation: while expressive models can be used for the base policy (e.g., to model the offline data distribution), the policy actually optimized and executed is a less expressive, one-step model. While a one-step model may suffice for simple simulation-based tasks, it is difficult to scale to larger base policies (e.g., VLAs) or more complex real-world settings, partly due to compounding errors between the teacher (the base policy) and the student (the distilled policy).

In this paper, we tackle the question:

*Can we develop a scalable offline RL approach*
*without relying on policy distillation or backpropagation through time?*

A natural alternative to policy gradients is rejection sampling: sample multiple action candidates from the base policy and choose the one with the highest value according to a learned value function (e.g., $Q$-function). However, existing rejection sampling methods suffer from two key limitations: the learned value function is not regularized, and value functions are typically limited to multilayer perceptron (MLP) networks. Standard approaches learn the value function from the offline dataset, resulting in $Q^{\pi_{\text{ref}}}$, the $Q$-function under the data-generating policy $\pi_{\text{ref}}$, which is not a provably optimal solution to the standard KL-regularized offline RL objective (Zhou et al., 2025a). Additionally, the $Q$-functions used in continuous state-action spaces are typically standard MLP networks. The primary method for improving the value function's expressivity is simply increasing the size of the value network, which requires more backpropagation through the additional layers in the neural network. Motivated by the recent success of generative models for computer vision and RL policies, we investigate how value learning can be improved by integrating flow matching.

Finally, we consider the third axis of scale—compute—and, in particular, how to leverage additional inference-time compute. Existing approaches to inference-time scaling typically use dynamics or world models for additional planning during inference, such as model predictive path integral control (Williams et al., 2017), model-based offline planning (Hafner et al., 2019; Argenson & Dulac-Arnold, 2020), planning with world models (Hafner et al., 2023), and Monte Carlo tree search (Chen et al., 2024). While promising, these methods either do not leverage expressive models, or they require learning and maintaining an auxiliary model of the environment, which can introduce additional sources of approximation error and scaling challenges.

The preceding limitations highlight a key gap in the scalability of existing offline RL approaches: although expressive generative models have been integrated into policies, the same level of expressivity has yet to be brought to value functions. In this paper, we bridge this gap through *Expressive Value Learning for Offline Reinforcement Learning* (EVOR): an approach for learning an optimal solution to the KL-regularized offline RL objective with *both* expressive policies *and* expressive value functions. EVOR achieves the following desiderata for scalable offline RL:

1. **EVOR avoids policy distillation and backpropagation through time during policy optimization.** EVOR avoids learning a new policy and instead optimizes the base policy through *inference-time policy extraction*. Unlike standard rejection sampling approaches, EVOR uses an optimal, regularized $Q$-function.

2. **EVOR learns an expressive, optimal $Q$-function via flow matching.** EVOR employs *flow-based temporal difference (TD) learning* to learn an optimal, regularized solution to the offline RL objective.

3. **EVOR enables inference-time scaling and regularization.** EVOR provides a natural mechanism for inference-time scaling: performing additional search, guided by the optimal value function, *without additional training*.

## 2 RELATED WORK

**Offline Reinforcement Learning.** Offline RL tackles the problem of learning a policy from a fixed dataset without additional environment interactions (Levine et al., 2020). In addition to the standard reward maximization goal of online RL, the key problem of offline RL is avoiding distribution shift between train-time (i.e., the offline dataset) and test-time (i.e., the learned policy's rollout). Numerous strategies have been proposed for the offline RL setting. A common approach is to employ behavior regularization, which forces the learned policy to stay close to the dataset via behavioral cloning or divergence penalties (Nair et al., 2020; Fujimoto & Gu, 2021; Tarasov et al., 2023a). Other approaches include in-distribution maximization (Kostrikov et al., 2022; Xu et al., 2023; Garg et al., 2023), dual formulations of RL (Lee et al., 2021; Sikchi et al., 2023), out-of-distribution detection (Yu et al., 2020; Kidambi et al., 2020; An et al., 2021; Nikulin et al., 2023), and conservative value estimation (Kumar et al., 2020). Farebrother et al. (2024); Nauman et al. (2025) propose training value functions via classification-based objectives, instead of the standard regression-based objectives. Rybkin et al. (2025) propose scaling laws for value-based reinforcement learning. Policies trained via offline

RL can subsequently be used for sample-efficient online RL in a procedure known as offline-to-online RL (Lee et al., 2021; Song et al., 2022; Nakamoto et al., 2023; Ball et al., 2023; Yu & Zhang, 2023; Ren et al., 2024b; Park et al., 2025b; Li et al., 2025).

**Offline Reinforcement Learning with Generative Models.** Standard offline RL approaches rely on Gaussian-based models in continuous state-action spaces. However, recent work has focused on representing policies via powerful sequence or generative models Chen et al. (2021); Janner et al. (2021; 2022); Wang et al. (2022); Ren et al. (2024a); Wu et al. (2024); Black et al. (2024); Park et al. (2025b); Espinosa-Dice et al. (2025), taking advantage of more powerful generative models like diffusion (Sohl-Dickstein et al., 2015; Ho et al., 2020; Song et al., 2021) and flow matching (Lipman et al., 2022; Liu et al., 2022; Lipman et al., 2024). These generative models are known to be more expressive than Gaussian-based models, enabling them to capture more complex, multi-modal distributions. Modeling complex distributions is particularly relevant to the offline RL setting, where the offline dataset may be composed of multiple data-generating policies of varying qualities. However, diffusion and flow models rely on an iterative sampling process that can be computationally expensive (Ding & Jin, 2023). To address this problem, some methods utilize a two-stage procedure to first train an expressive generative model on the offline dataset, and then distill it into a one-step model that is then used for policy optimization (Ding & Jin, 2023; Chen et al., 2023; Meng et al., 2023; Park et al., 2025b). Espinosa-Dice et al. (2025) propose an approach for avoiding both distillation and extensive backpropagation through time by leveraging shortcut models for flexible inference, but rely on a standard, Gaussian-based value function. Additionally, generative models have been used for plan generation in offline RL (Zheng et al., 2023) and energy-guided flow and diffusion models, incorporating reward feedback in the flow and diffusion training (Zhang et al., 2025). Farebrother et al. (2025) propose integrating flow matching with Bellman-style updates for successor representation learning.

**Inference-Time Scaling in Offline Reinforcement Learning.** Inference-time scaling in reinforcement learning often takes the form of leveraging dynamics or world models for additional planning during inference. Approaches include model predictive control (Richalet et al., 1978; Hansen et al., 2022), model predictive path integral control (Williams et al., 2015; 2017; Gandhi et al., 2021), model-based offline planning (Hafner et al., 2019; Argenson & Dulac-Arnold, 2020), sequence modeling (Janner et al., 2021; 2022; Kong et al., 2024), planning with world models (Hafner et al., 2023), and Monte Carlo tree search (Chen et al., 2024). Additional approaches include applying rejection sampling to the learned value function at inference-time (Chen et al., 2022; Fujimoto et al., 2019; Ghasemipour et al., 2021; Hansen-Estruch et al., 2023; Park et al., 2024b; Nakamoto et al., 2024) or using the gradient of the learned value function to adjust actions at inference-time (Park et al., 2024b). Generative models like flow matching and diffusion models naturally support a form of sequential scaling by increasing the number of steps in the iterative sampling process (Ho et al., 2020; Song et al., 2020; Liu et al., 2022; Lipman et al., 2022). Espinosa-Dice et al. (2025) takes advantage of flexibility in the number of denoising steps used when sampling actions from the policy. However, existing approaches do not leverage generative models for value learning like EVOR. By leveraging more expressive models for value learning, EVOR can better take advantage of larger, more complex offline datasets.

## 3 BACKGROUND

**Markov Decision Process.** We consider a finite-horizon Markov decision process (MDP) $(\mathcal{X}, \mathcal{A}, P, r, H)$, where $\mathcal{X}$ is the state space, $\mathcal{A}$ is the action space, $P$ is the transition function, $r : \mathcal{X} \times \mathcal{A} \to [0, 1]$ is the reward function, and $H$ is the MDP's horizon (Puterman, 2014). An offline dataset $\mathcal{D} = \{(x_h, a_h, r_h, x_{h+1})\}$ is collected under some unknown reference policy $\pi_{\text{ref}}$, which could be multi-modal and sub-optimal. In the offline RL setting, we do not assume access to environment interactions.

**Offline Reinforcement Learning.** The offline RL objective is generally expressed as a combination of a policy optimization term and a regularization term, such that

$$\underset{\pi \in \Pi}{\operatorname{argmax}} \quad \underbrace{J_{\mathcal{D}}(\pi)}_{\texttt{Policy Optimization}} - \underbrace{\eta \operatorname{Reg}(\pi, \pi_{\text{ref}})}_{\texttt{Regularization}} \tag{1}$$

where $J_{\mathcal{D}}(\pi)$ is the expected return over offline dataset $\mathcal{D}$, $\pi_{\text{ref}}$ is the unknown data-generating policy, and $\operatorname{Reg}(\pi, \pi_{\text{ref}})$ is a regularization term (Espinosa-Dice et al., 2025). The regularization term generally takes the form of a divergence measure between $\pi$ and $\pi_{\text{ref}}$, with KL divergence being most common. The offline RL objective can be expressed as the

soft value of a policy subject to KL regularization:

$$V^{\pi,\eta} = \mathbb{E}_\pi \left[ \sum_{h=1}^{H} r(x_h, a_h) - \eta \mathrm{KL} \left( \pi(x_h) \| \pi_{\mathrm{ref}}(x_h) \right) \right], \tag{2}$$

where the expectation is over a random trajectory $(x_1, a_1, \ldots, x_H, a_H)$ sampled according to $\pi$ and the KL divergence is $\mathrm{KL}(p\|q) = \mathbb{E}_{z\sim p}\left[\log\left(p(z)/q(z)\right)\right]$ (Zhou et al., 2025a). The objective is to learn the optimal, regularized policy $\pi^\star = \mathrm{argmax}_{\pi\in\Pi} V^{\pi,\eta}$. Ziebart et al. (2008) showed that

$$V_{H+1}^{\star,\eta}(x) = 0, \tag{3}$$

$$Q_h^{\star,\eta}(x, a) = r(x, a) + \mathbb{E}_{x'\sim P_h(x,a)} \left[ V_{h+1}^{\star,\eta}(x') \right], \tag{4}$$

$$\pi^{\star,\eta}(a|x) \propto \pi_{\mathrm{ref}}(a|x) \exp\left(\eta^{-1} Q_h^{\star,\eta}(x, a)\right), \tag{5}$$

$$V_h^{\star,\eta}(x) = \eta \ln \mathbb{E}_{a\sim\pi_{\mathrm{ref}}(\cdot|x)}(x) \left[ \exp\left(\eta^{-1} Q_h^{\star,\eta}(x, a)\right) \right]. \tag{6}$$

For convenience, we drop the $\eta$ superscript when clear from context.

**Reward-To-Go.** We define the reward-to-go under the unknown data-generating policy $\pi_{\mathrm{ref}}$, starting at state $x$ and taking action $a$, as

$$Z(x, a) := \sum_{h=0}^{H} r(x_h, a_h), \quad x_0 = x, \ a_0 = a, \ x_{h+1} \sim P_h(\cdot \mid x_h, a_h), \ a_{h+1} \sim \pi_{\mathrm{ref}}(\cdot \mid x_{h+1}), \tag{7}$$

We define $R(\cdot \mid x, a)$ as the law of the random variable $Z(x, a)$, so $R(\cdot \mid x, a) \overset{D}{=} Z(x, a)$. In other words, $R(\cdot \mid x, a)$ is the distribution of rewards-to-go under $\pi_{\mathrm{ref}}$, starting at state $x$ and taking action $a$. We can thus define

$$\pi^{Z,\eta}(a|x) \propto \pi_{\mathrm{ref}}(a|x) \mathbb{E}_{z\sim Z(x,a)} \left[ \exp\left(z/\eta\right) \right]. \tag{8}$$

We can also define $R^\pi(\cdot \mid x, a)$ as the distribution of rewards-to-go under a policy $\pi$.

**Flow Matching.** We define flow matching (Lipman et al., 2022; Liu et al., 2022; Lipman et al., 2024) as follows. Let $p(x) \in \Delta(\mathbb{R}^d)$ be a data distribution. Given a vector field $v_t$, we construct its corresponding flow, $\phi : [0, 1] \times \mathbb{R}^d \to \mathbb{R}^d$, by the ordinary differential equation (ODE)

$$\frac{d}{dt}\phi_t(x) = v_t(\phi_t(x)) \tag{9}$$

$$\phi_0(x) = x \tag{10}$$

We employ Lipman et al. (2024)'s flow matching, which is based on linear paths and uniform time sampling, such that the training objective is

$$\min_\theta \mathbb{E}_{\substack{x^0\sim\mathcal{N}(0,I_d) \\ x^1\sim p(x) \\ t\sim U[0,1]}} \left[ \| v_\theta(t, x^t) - (x^1 - x^0) \|_2^2 \right] \tag{11}$$

where $x^t = (1 - t)x^0 + tx^1$ is the linear interpolation between $x^0$ and $x^1$.

## 4 EXPRESSIVE VALUE LEARNING FOR OFFLINE REINFORCEMENT LEARNING

In this section, we tackle the question:

*How do we learn an optimal, expressive value function for the KL-regularized offline RL objective?*

Our key insight is to integrate the power of flow matching in TD learning through a distributional approach, which we refer to as *Expressive Value Learning for Offline Reinforcement Learning* (EVOR).

### 4.1 LEARNING A BASE POLICY THROUGH FLOW MATCHING

In offline reinforcement learning, we aim to learn or fine-tune a policy from a dataset collected under an unknown data-generating policy $\pi_{\mathrm{ref}}$. Depending on the setting, we either have access to a base policy $\pi_{\mathrm{base}}$ (e.g., a pre-trained generalist model) or we must learn the policy from scratch. Both settings are compatible with our approach, and we first show how a base policy can be learned in the setting when no initial base policy is provided.

**Algorithm 1:** EVOR Training via Flow-Based TD Learning

---

**Input:** Offline dataset $\mathcal{D}$

**while** not converged **do**

    Sample $(x, a^1, x', r) \sim \mathcal{D}$                                    `# Parallelize batch`

    ▷ **Base Policy Update via Flow Matching**

    $a^0 \sim \mathcal{N}(0, I_d),\ t \sim \text{Unif}(0,1)$                       `# Sample noise and time`

    $a^t \leftarrow (1-t)a^0 + ta^1$                             `# Noise action`

    $\phi \leftarrow \nabla_\phi \| v_\phi(a^t, t \mid x) - (a^1 - a^0) \|^2$              `# Update actor`

    ▷ **Reward Model Update via Flow-Based TD Learning**

    $z^0 \sim \mathcal{N}(0, I_d),\ z^1 \sim R(\cdot \mid x, a),\ t \sim \text{Unif}(0,1)$     `# Sample noise, reward-to-go, time`

    $z^t \leftarrow (1-t)z^0 + tz^1$                                `# Noise reward-to-go`

    $a' \sim \pi_{\text{base}}(\cdot \mid x')$                           `# Sample action from base policy`

    $s_{\text{target}} \leftarrow r(x, a) + \overline{s}_\theta(z^t, t \mid x', a')$       `# Flow-matching target`

    $\theta \leftarrow \nabla_\theta \| s_\theta(z^t, t \mid x, a) - \text{stopgrad}(s_{\text{target}}) \|^2$    `# Update critic`

---

**Base Policy Objective.** In the setting where a starting base policy is not known, we train a policy $\pi_{\text{base}}$ that predicts actions via behavioral cloning (Pomerleau, 1988) on the offline dataset's state-action pairs. By formulating the objective as a supervised learning problem rather than a more complex RL procedure, we can employ any generative model to learn the base policy. We use flow matching here, such that the loss is given by:

$$\mathcal{L}_{\text{BC}}(\phi) = \mathbb{E}_{\substack{(x,a^1)\sim\mathcal{D},\, a^0\sim\mathcal{N} \\ t\sim\text{Unif}(0,1)}} \left[ \Big\| \underbrace{v_\phi(a^t, t \mid x)}_{\texttt{Velocity Prediction}} - \underbrace{(a^1 - a^0)}_{\texttt{Velocity Target}} \Big\|^2 \right] \tag{12}$$

where $a^0$ represents a fully noised action (i.e., noise sampled from a Gaussian) and $a^1$ represents a real action (i.e., action sampled from the offline data $\mathcal{D}$). Leveraging an expressive model like flow matching enables $\pi_{\text{base}}$ to model complex action distributions and multi-modal offline data. Through Equation 12, we learn a base policy $\pi_{\text{base}} \approx \pi_{\text{ref}}$, subject to finite sample and optimization errors.

## 4.2 EXPRESSIVE VALUE LEARNING VIA FLOW-BASED TD LEARNING

Next, we turn to the problem of learning an expressive value function. Rather than use standard value learning methods, our key insight is to integrate flow matching into distributional TD learning, thus enabling us to leverage the expressivity of flow matching with the variance reduction and improved credit assignment of TD learning.

Intuitively, we can think of flow matching as a method of transporting samples from a prior distribution (e.g., samples from a Gaussian) to a target distribution (i.e., the data distribution). For EVOR, we set the target distribution to the distribution of rewards-to-go under $\pi_{\text{ref}}$, denoted by $R(\cdot \mid x, a)$, which is a distribution we have samples from the offline dataset. In the remainder of the subsection, we explain the *flow-based temporal difference (TD) learning* objective and high-level intuition behind it. Its formal derivation can be found in Appendix A.

**Distributional Bellman.** Recall that TD learning uses the Bellman equation to learn a value function by constructing a bootstrap target (i.e., the right-hand side (RHS) of the Bellman equation) (Bellman, 1966; Sutton & Barto, 1998), such that

$$Q(x, a) = r(x, a) + \gamma \mathbb{E}_{P,\pi} Q(X', A'). \tag{13}$$

Notably, the Bellman equation also holds under distributions (Jaquette, 1973; Sobel, 1982; White, 1988; Bellemare et al., 2017), such that

$$\underbrace{Z(x, a)}_{\texttt{LHS of Distributional Bellman}} \overset{D}{=} \overbrace{r(x, a) + \gamma Z(X', A')}^{\texttt{RHS of Distributional Bellman}} \tag{14}$$

where $Z(X', A')$ denotes the random return.

---

**Algorithm 2:** `EVOR` Inference via $Q_\theta^\star$ Reweighting

---

**Input:** State $x$; number of action candidates $N_\pi$; number of reward-to-go samples $N$; temperatures $\tau_R, \tau_Q$

▷ **Sample candidate actions and reward-to-go's**

$\{a^{(i)}\}_{i=1}^{N_\pi} \sim \pi_{\text{base}}(\cdot \mid x)$                        `# Sample` $N_\pi$ `candidate actions`

$\{r^{(i,j)}\}_{j=1}^{N} \sim R_\theta(\cdot \mid x, a^{(i)})$                     `# Sample` $N$ `reward-to-go's`

▷ **Sample average and apply softmax**

$Q_\theta^\star(x, a^{(i)}) \leftarrow \tau_R \operatorname{LogSumExp}_{r \in \{r^{(i,j)}\}_{j=1}^{N}} (r/\tau_R)$          `# Sample average` $Q_\theta^\star$

$a^\star \sim \operatorname{softmax}_{a \in \{a^{(i)}\}_{i=1}^{N_\pi}} \left( Q_\theta^\star(x, a)/\tau_Q \right)$        `# Softmax over action candidates`

**return** $a^\star$

---

**Flow-Based TD Objective.** Flow matching learns to transport a prior distribution into a target data distribution. The distributional Bellman equation provides two distributions on either side of the equation. To construct a flow-based TD objective, we set the RHS of the distributional Bellman equation as the target distribution, and match the velocities between the LHS and RHS distributions.

More specifically, we learn a conditional flow model $s_\theta(\cdot \mid x, a, t)$ that transports base noise sampled from a Gaussian, $z_{x,a}^0 \sim \mathcal{N}(0, I_d)$, to a terminal variable $z_{x,a}^1 \sim R_\theta(\cdot \mid x, a)$, such that the learned distribution $R_\theta(\cdot \mid x, a) \approx R^{\pi_{\text{base}}}(\cdot \mid x, a)$, where $R^{\pi_{\text{base}}}(\cdot \mid x, a)$ is the distribution of rewards-to-go of $\pi_{\text{base}}$. The bootstrap target is given by

$$s_{\text{target}} := r(x, a) + \gamma \mathbb{E}_{a' \sim \pi_{\text{base}}(\cdot \mid x')} \bar{s}_\theta(z^t \mid x', a', t), \tag{15}$$

and the loss is given by

$$\mathcal{L}_{\text{FlowTD}}(\theta) = \underbrace{\mathbb{E}_{(x,a,r,x') \sim \mathcal{D}}}_{\text{Dataset's State-Action-Reward}} \overbrace{\mathbb{E}_{z^1 \sim \bar{R}_\theta(\cdot \mid x,a)}}^{\text{Sample Reward-To-Go}} \underbrace{\mathbb{E}_{t \sim \text{Unif}(0,1)}}_{\text{Sample Time}} \left\| \overbrace{s_\theta(z^t \mid x, a, t)}^{\text{Velocity Prediction of LHS}} - \underbrace{\text{stopgrad}(s_{\text{target}})}_{\text{Velocity Target of RHS}} \right\|_2^2. \tag{16}$$

We sample a state-action-reward-next-state tuple from the offline data $(x, a, r, x') \sim \mathcal{D}$, a time $t \sim \text{Unif}(0, 1)$, and the next action from the base policy $a' \sim \pi_{\text{base}}(\cdot \mid x')$. We construct a linear interpolant $z^t = (1 - t)z^0 + tz^1$ by sampling a reward-to-go $z^1 \sim R(\cdot \mid x, a)$ and a noise sample $z^0 \sim \mathcal{N}(0, I_d)$. The reward-to-go sample $z^1$ can be sampled from the dataset $\mathcal{D}$ or a target version of the learned reward model $\bar{R}_\theta(\cdot \mid x, a)$. We sample a reward-to-go from $R_\theta(\cdot \mid x, a)$ using the standard forward Euler method applied on the learned flow model $s_\theta(\cdot \mid x, a, t)$.

**Flow-Based TD Target.** The velocity target in Equation 15 represents a flow field that generates samples from the RHS of the distributional Bellman (Equation 14), $r(x, a) + Z(X', A')$. The RHS of the distributional Bellman (i.e., the target distribution in flow matching) corresponds to the reward-to-go distribution translated by the one-step reward $r(x, a)$. Under flow matching, such a translation shifts every particle in the flow trajectory by a constant amount. The vector field specifies the instantaneous rate of change of particle positions, so adding a constant shift to all trajectories increases the velocity everywhere by the same constant. Consequently, the target $s_{\text{target}} = r(x, a) + \gamma \mathbb{E}_{a' \sim \pi_{\text{base}}(\cdot \mid x')} \bar{s}_\theta(z^t \mid x', a', t)$ is the one-step reward shift and the expected next-state flow under $\pi_{\text{base}}$.

Next, we consider the question:

*How do we obtain an optimal value function from the reward model?*

Building on results from Ziebart et al. (2008) and Zhou et al. (2025a), we show that the optimal, regularized $Q$-function can be obtained using the learned reward model.

Table 1: **EVOR**'s Overall Performance. EVOR achieves the best overall performance across six environments, for a total of 27 unique tasks in the OGBench (Park et al., 2024a) and D4RL task suites (Fu et al., 2020). Results are averaged over five seeds per task for OGBench and three seeds per task for D4RL, with standard deviations reported. IQL's D4RL results are reported from prior work Tarasov et al. (2023b). The full results are reported in Appendix B.

| Task Category | IQL | QC-1 | QC-5 | EVOR |
|---|---|---|---|---|
| OGBench antmaze-large-navigate-singletask (5 tasks) | $45_{\pm 5}$ | $9_{\pm 6}$ | $7_{\pm 2}$ | $\mathbf{50}_{\pm 4}$ |
| OGBench antmaze-large-stitch-singletask (5 tasks) | $\mathbf{30}_{\pm 6}$ | $9_{\pm 5}$ | $8_{\pm 4}$ | $15_{\pm 1}$ |
| OGBench cube-double-play-100M-singletask (5 tasks) | $55_{\pm 6}$ | $55_{\pm 5}$ | $57_{\pm 19}$ | $\mathbf{81}_{\pm 2}$ |
| OGBench pointmaze-medium-naviate-singletask (5 tasks) | $\mathbf{99}_{\pm 0}$ | $91_{\pm 10}$ | $\mathbf{99}_{\pm 0}$ | $\mathbf{99}_{\pm 0}$ |
| OGBench scene-play-sparse-singletask (5 tasks) | $48_{\pm 6}$ | $47_{\pm 4}$ | $\mathbf{83}_{\pm 4}$ | $87_{\pm 6}$ |
| D4RL locomotion (2 tasks) | $36$ | $36_{\pm 16}$ | $21_{\pm 7}$ | $\mathbf{39}_{\pm 9}$ |

**Theorem 1** (Optimal Regularized Value Functions (Zhou et al., 2025a)). *Under deterministic transitions, the optimal value and Q-functions are given by*

$$V_h^{\star,\pi}(x_h) = \eta \ln \mathbb{E}_{\pi_{\text{ref}}}\left[\exp\left(\eta^{-1}\sum_{t\geq h}^{H} r(x_t, a_t)\right) \,\bigg|\, x_h\right], \tag{17}$$

$$Q_h^{\star,\pi}(x_h, a_h) = \eta \ln \mathbb{E}_{\pi_{\text{ref}}}\left[\exp\left(\eta^{-1}\sum_{t\geq h}^{H} r(x_t, a_t)\right) \,\bigg|\, x_h, a_h\right]. \tag{18}$$

Using Theorem 1, we can express the optimal, regularized $Q$-function as a function of $\pi_{\text{ref}}$'s reward-to-go distribution $R(\cdot \mid x, a)$, such that

$$Q_h^{\star}(x_h, a_h) = \eta \ln \mathbb{E}_{z \sim R_h(\cdot | x_h, a_h)} \exp\left(\eta^{-1} z\right) \tag{19}$$

Through the flow-based TD learning objective (Equation 16), we learn a reward model $R_\theta(\cdot \mid x, a) \approx R^{\pi_{\text{base}}}(\cdot \mid x, a)$, where $\pi_{\text{base}} \approx \pi_{\text{ref}}$ if the base policy is learned well. In practice, we can approximate the expectation via sample averaging, and the full training procedure is shown in Algorithm 1. While the assumption of deterministic dynamics can be strong in certain settings, it is frequently imposed in the analysis of offline RL algorithms (Edwards et al., 2020; Ma et al., 2022; Schweighofer et al., 2022; Park et al., 2023; Ghosh et al., 2023; Wang et al., 2023; Karabag & Topcu, 2023; Park et al., 2024a;c), and we only use it here to derive the optimal expression for the $Q$-function. We then empirically validate our results on recent offline RL benchmarks in Section 5.

### 4.3 INFERENCE-TIME POLICY EXTRACTION, REGULARIZATION, AND SCALING

EVOR's training procedure focuses on learning an expressive value function, and it trains the base policy via flow matching on the offline dataset, leading to the natural question:

> *How does EVOR optimize the base policy beyond the offline dataset*
> *without distillation or backpropagation through time?*

**Inference-Time Policy Extraction.** Instead of learning a new policy during training, EVOR performs *inference-time policy extraction* using the learned distributional reward model. A common approach to inference-time policy extraction is to perform rejection sampling with the learned value or $Q$-function (Fujimoto et al., 2019; Ghasemipour et al., 2021; Gui et al., 2024; Nakamoto et al., 2024). Given a state $x$, sample actions independently from the base policy $a_1, a_2, \ldots, a_N \sim \pi_{\text{base}}(\cdot \mid x)$, and select the action with the largest $Q$ value, such that

$$\underset{a \in \{a_1, a_2, \ldots, a_N\}}{\operatorname{argmax}} Q(x, a) \tag{20}$$

However, using the $Q$-function trained on the offline dataset $\mathcal{D}$, which is generated by $\pi_{\text{ref}}$, is not an optimal solution to the KL-regularized offline RL objective, and optimizing it may lead to distribution shift and poor performance at test-time (Zhou et al., 2025a). Instead, we utilize our expression for the *optimal Q-function*,

$$Q^{\star}(x, a) = \eta \ln \mathbb{E}_{r \sim R(\cdot | x, a)} \exp(r/\eta), \tag{21}$$

where $R$ is the conditional distribution of rewards-to-go under $\pi_{\text{ref}}$. In practice, we approximate the expectation via sample averaging, and we can construct a softmax over the $Q^{\star}$ values, as shown in Algorithm 2.

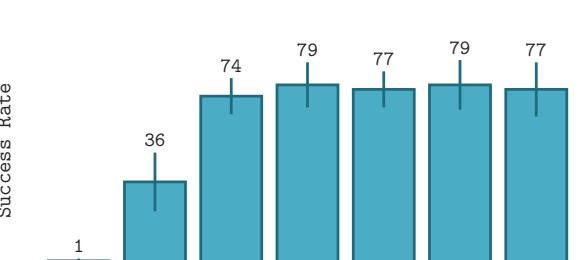

Figure 1: **EVOR's Inference-Time Scaling.** EVOR can perform inference-time scaling by increasing the number of action candidates $N_\pi$, performing greater search at inference time with the expressive value function. Leveraging greater inference-time compute results in better performance, up to a saturation point. Results are averaged over three seeds per task, with standard deviations reported.

**Inference-Time Regularization and Scaling.** EVOR's formulation provides a natural mechanism for inference-time regularization and scaling. Since actions are sampled from the base policy, running EVOR with varying temperatures $\tau_R$ and $\tau_Q$ controls the strength of regularization and policy optimization. Increasing $N_\pi$ corresponds to performing additional test-time search, while decreasing it enables faster inference under smaller compute budgets. Crucially, these parameters can be varied at test-time *without retraining*, allowing for both inference-time scaling and regularization.

## 5 EXPERIMENTAL RESULTS

In this section, we investigate the performance of EVOR, focusing on the following question:

*What is the benefit of expressive value learning?*

### 5.1 EXPERIMENTAL SETUP

**Environments and Tasks.** We follow the experimental setup of prior works leveraging the OGBench task suite (Park et al., 2024a; 2025b; Espinosa-Dice et al., 2025; Li et al., 2025), specifically evaluating EVOR on locomotion and manipulation robotics tasks. We describe the full implementation details in Appendix D.

**Baselines.** Rather than comparing to all of the existing offline RL algorithms benchmarked on OGBench, we aim to isolate the effect of expressive value learning in order to demonstrate its benefit specifically. Thus, we compare to $Q$-chunking (QC, Li et al. (2025)), a recent offline RL algorithm that is closest to EVOR. Like EVOR, QC learns a base policy via flow matching and extracts an optimized policy via rejection sampling. The key difference between QC and EVOR is how the value function is learned—the exact difference we aim to isolate. QC can employ action chunking in both its policy and value function, and we compare EVOR to both QC with (QC-5) action chunking and without it (QC-1). We select the action chunk length (5) based on Li et al. (2025)'s recommendation.

Additionally, we add Implicit Q-Learning (IQL) (Kostrikov et al., 2022) as an additional baseline, which presents a separate approach for learning a value function, specifically through expectile regression. IQL then extracts the policy through advantage-weighted regression (Peters & Schaal, 2007; Wang et al., 2018; Peng et al., 2019; Nair et al., 2020).

**Evaluation.** For a fair comparison, we use the same network size, number of gradient steps, and discount factor across all algorithms, following Park et al. (2025b); Espinosa-Dice et al. (2025). Moreover, we use the official QC implementation and its parameters. We bold values at 95% of the best performance in tables.

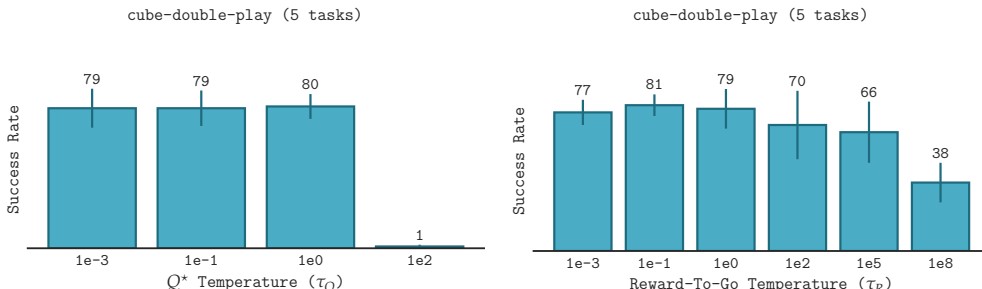

Figure 2: **Ablation Over EVOR's Evaluation Parameters.** EVOR uses the same training parameters for all environments in this paper. However, we investigate the effect of varying the temperature parameters $\tau_R$ and $\tau_Q$ at inference-time on the performance of EVOR. As $\tau_Q$ decreases, the action selection becomes more greedy, while as $\tau_Q$ increases, the action selection becomes more regularized. Set to a high value, EVOR becomes equivalent to the base policy (i.e., the performance with $N_\pi = 1$). Results are averaged over three seeds per task, with standard deviations reported.

## 5.2 EXPERIMENTAL RESULTS

**Q: What is EVOR's overall performance?**

Across six environments and 27 distinct tasks, EVOR achieves the best overall performance compared to the baselines. Environment-level aggregation results are shown in Table 1, with full task-level results in Appendix B.

**Q: Does using expressive models for value learning improve performance?**

Yes. As shown in Table 1, EVOR 's expressive value learning approach consistently outperforms standard value learning, including those employing action chunking (QC-5), as well as expectile regression-based value learning methods (IQL). The results suggest that expressive value learning provides performance benefits in the settings considered.

**Q: How can EVOR take advantage of greater inference-time compute?**

As shown in Figure 1, with greater inference-time compute, EVOR can evaluate more action candidates ($N_\pi$), leading to improved performance (up to a saturation point). We present the full results for inference-time scaling in Appendix C.

**Q: How can EVOR perform inference-time regularization?**

As shown in Figure 2, by varying the temperature parameters $\tau_R$ and $\tau_Q$, EVOR can vary the level of regularization to the base policy compared to the level of policy optimization. As $\tau_Q$ decreases, the action selection becomes more greedy; as $\tau_Q$ increases, the action selection becomes more regularized to the base policy $\pi_{\text{base}}$ (i.e., the performance of EVOR with $N_\pi = 1$).

**Q: What training parameters must EVOR tune per environment?**

A key advantage of EVOR is that it uses the same training and evaluation parameters for all environments in Table 1, despite the environments spanning distinct locomotion and manipulation tasks. In contrast, policy gradient-based offline RL algorithms generally tune parameters per environment (Park et al., 2024a; 2025b; Espinosa-Dice et al., 2025). We present an ablation study of evaluation parameters in Appendix C.

**Q: Does rejection sampling-based policy extraction outperform reparameterized policy gradients?**

We do not consider that question in this work. The purpose of this paper is to investigate scalable methods for expressive value learning in offline RL. In our empirical results, we isolate the effect of expressive value learning by using a consistent policy extraction method (rejection sampling). We acknowledge that rejection sampling may not be the most effective policy extraction scheme, as argued by Park et al. (2024b). However, unlike policy gradient methods, rejection sampling does not require backpropagation through time or distillation—both key bottlenecks in scaling offline RL that this paper seeks to avoid.

## 6 DISCUSSION

In summary, EVOR is a scalable approach to offline reinforcement learning that integrates *both* expressive policies *and* expressive value learning. EVOR learns an optimal solution to the KL-regularized offline RL objective, which enables inference-time policy extraction without model distillation or backpropagation through time. Furthermore, EVOR can perform inference-time scaling by performing greater search, guided by the expressive value function, and it can adjust the level of regularization to the base policy without retraining. In this paper, we focus on scalable offline RL by avoiding distillation and backpropagation through time, leading us to rejection sampling against an expressive value function. However, as noted by Park et al. (2024b), reparameterized policy gradients are an effective policy extraction technique. Future work may explore how EVOR's expressive value learning could be integrated with policy gradient techniques and action chunking in a scalable manner.

### REPRODUCIBILITY STATEMENT

We have taken several steps to ensure the reproducibility of our work. All code necessary to reproduce our experiments, along with instructions for installation and execution, is included in the supplementary materials as an anonymized repository. Detailed descriptions of the experimental setup and parameters are provided in Appendix D. The environments and datasets used in our experiments are publicly available. Together, these resources enable independent verification of our findings. We employ LLMs to aid and polish writing based on drafts that we wrote.

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

## A  FLOW-BASED TEMPORAL DIFFERENCE LEARNING

We restate the flow-based TD objective and describe its derivation.

**Distributional Reinforcement Learning.**   TD learning uses the Bellman equation to learn a value function by constructing a bootstrap target (i.e., the right-hand side (RHS) of the Bellman equation) (Bellman, 1966; Sutton & Barto, 1998), such that

$$Q(x,a) = r(x,a) + \gamma \mathbb{E}_{P,\pi} Q(X', A'). \tag{22}$$

The Bellman equation also -holds under distributions (Jaquette, 1973; Sobel, 1982; White, 1988; Bellemare et al., 2017), such that

$$\underbrace{Z(x,a)}_{\text{LHS of Distributional Bellman}} \overset{D}{=} \overbrace{r(x,a) + \gamma Z(X', A')}^{\text{RHS of Distributional Bellman}} \tag{23}$$

where $Z(X', A')$ denotes the random return.

**Goal.**   At a high level, flow matching learns to transport a known prior distribution into a target data distribution. To construct a flow-based TD objective, we set the RHS of the distributional Bellman equation as the target distribution, and match the velocities between the LHS and RHS distributions. We learn a conditional flow model $s_\theta(\cdot \mid x, a, t)$ that transports base noise $Y_{x,a}(0) \sim \mathcal{N}(0, I_d)$ to a terminal variable $Y_{x,a}(1) \sim R_\theta(\cdot \mid x, a)$, such that the distribution $R_\theta(\cdot \mid x, a) \approx R(\cdot \mid x, a)$.

**Conditional Flow Model.**   We learn a conditional velocity field $s_\theta(y \mid x, a, t)$ that defines the ODE

$$\frac{d}{dt} Y_{x,a}(t) = s_\theta(Y_{x,a}(t) \mid x, a, t), \qquad Y_{x,a}(0) \sim p_0. \tag{24}$$

Solving (i.e., "running") this ODE from $t = 0$ to $t = 1$ is done by integration, giving the terminal random variable

$$Y_{x,a}(1) = Y_{x,a}(0) + \int_0^1 s_\theta(Y_{x,a}(\tau) \mid x, a, \tau) d\tau. \tag{25}$$

Let $R_\theta(\cdot \mid x, a)$ denote the induced terminal distribution. Our goal is to learn $R_\theta(\cdot \mid x, a) \approx R(\cdot \mid x, a)$.

**Distributional Bellman.**   By the definition of discounted reward-to-go,

$$Z(x,a) \overset{D}{=} r(x,a) + \gamma Z(X', A'), \tag{26}$$

where $X' \sim P(\cdot \mid x, a)$, $A' \sim \pi_{\text{base}}(\cdot \mid X')$, and $Z(X', A') \sim R(\cdot \mid X', A')$. Equivalently, we can say

$$R(\cdot \mid x, a) = \mathcal{L}\left(r(x,a) + \gamma Z'\right), \quad Z' \sim R(\cdot \mid X', A'), \tag{27}$$

where $\mathcal{L}$ is the law of the random variable. Taking expectation of Equation 26 yields

$$\mathbb{E}_{Z \sim R(\cdot \mid x, a)}[Z] = r(x,a) + \gamma \mathbb{E}_{X' \sim P(\cdot \mid x, a)} \mathbb{E}_{A' \sim \pi_{\text{base}}(\cdot \mid X')} \mathbb{E}_{Z' \sim R(\cdot \mid X', A')}[Z']. \tag{28}$$

**Flow Integral and Expectation.**   Going back to the ODE solution, we have

$$Y_{x,a}(1) = Y_{x,a}(0) + \int_0^1 s_\theta(Y_{x,a}(\tau) \mid x, a, \tau) d\tau. \tag{29}$$

Taking expectation first and then applying Fubini's theorem, we have

$$\mathbb{E}\left[Y_{x,a}(1) \mid x, a\right] = \mathbb{E}\left[Y_{x,a}(0)\right] + \mathbb{E}\left[\int_0^1 \left[s_\theta(Y_{x,a}(\tau) \mid x, a, \tau)\right] d\tau \mid x, a\right] \tag{30}$$

$$= \mathbb{E}\left[Y_{x,a}(0)\right] + \int_0^1 \mathbb{E}\left[s_\theta(Y_{x,a}(\tau) \mid x, a, \tau) \mid x, a\right] d\tau. \tag{31}$$

By definition of $p_0$ being zero mean, $\mathbb{E}[Y_{x,a}(0)] = 0$, leaving us with:

$$\mathbb{E}\left[Y_{x,a}(1) \mid x, a\right] = \int_0^1 \mathbb{E}\left[s_\theta(Y_{x,a}(\tau) \mid x, a, \tau) \mid x, a\right] d\tau. \tag{32}$$

If we perform flow matching well, such that $R_\theta(\cdot \mid x, a) \approx R(\cdot \mid x, a)$ (subject to finite sample and optimization errors), then

$$\int_0^1 \mathbb{E}\left[s_\theta(Y_{x,a}(\tau) \mid x, a, \tau) \mid x, a\right] d\tau = r(x, a) + \gamma\mathbb{E}_{X', A'}\left[\int_0^1 \mathbb{E}\left[s_\theta(Y_{X',A'}(\tau) \mid X', A', \tau) \mid X', A'\right] d\tau\right]. \tag{33}$$

**Flow-Based TD Objective.** Equation 33 specifies an integral-level condition. To construct a tractable training objective, we add a pointwise (i.e. translation) condition, specifying that the equality hold for all $t \in [0, 1]$, such that

$$\mathbb{E}\left[s_\theta(Y_{x,a}(t) \mid x, a, t) \mid x, a\right] = r(x, a) + \gamma\mathbb{E}_{X', A'}\mathbb{E}\left[s_\theta(Y_{X',A'}(t) \mid X', A', t) \mid X', A'\right], \quad \forall t \in [0, 1]. \tag{34}$$

The pointwise condition is stronger than the integral-level condition, but it is natural for rectified flows, where the velocity target represents a flow field that generates samples from the RHS of the distributional Bellman (Equation 26), $r(x, a) + Z(X', A')$. The RHS of the distributional Bellman (i.e., the target distribution in flow matching) corresponds to the reward-to-go distribution translated by the one-step reward $r(x, a)$. Under flow matching, such a translation shifts every particle in the flow trajectory by a constant amount. Since we employ the linear path variant of flow matching, where particles move along straight-line interpolants, the Bellman translation $r(x, a)$ corresponds to an additive shift in the target velocity field. The vector field specifies the instantaneous rate of change of particle positions, so adding a constant shift to all trajectories increases the velocity everywhere by the same constant. Consequently, the target $s_{\text{target}} = r(x, a) + \gamma\mathbb{E}_{a' \sim \pi_{\text{base}}(\cdot | x')}\bar{s}_\theta(z^t \mid x', a', t)$ is the one-step reward shift and the expected next-state flow under $\pi_{\text{base}}$. In practice, the pointwise condition from Equation 34 is enforced via uniformly sampling $t$ and minimizing the mean squared error (MSE).

Putting this all together, we have the flow-based TD loss:

$$\mathcal{L}_{\text{FlowTD}}(\theta) = \underbrace{\mathbb{E}_{(x,a,r,x') \sim \mathcal{D}}}_{\text{Dataset's State-Action-Reward}} \overbrace{\mathbb{E}_{z^1 \sim R_{\bar{\theta}}(\cdot | x, a)}}^{\text{Sample Reward-To-Go}} \underbrace{\mathbb{E}_{t \sim \text{Unif}(0,1)}}_{\text{Sample Time}} \left\| \overbrace{s_\theta(z^t \mid x, a, t)}^{\text{Velocity Prediction of LHS}} - \underbrace{\text{target}(x, a, z^t, t)}_{\text{Velocity Target of RHS}} \right\|_2^2, \tag{35}$$

where

$$\text{target}(x, a, z^t, t) := r(x, a) + \gamma\mathbb{E}_{a' \sim \pi_{\text{base}}(\cdot | x')} s_{\bar{\theta}}(z^t \mid x', a', t). \tag{36}$$

We sample a state-action-reward-next-state tuple $(x, a, r, x') \sim \mathcal{D}$ from the offline data, a time $t \sim \text{Unif}(0, 1)$, and the next action from the base policy $a' \sim \pi_{\text{base}}(\cdot \mid x')$. We construct an interpolant $z^t = (1 - t)z^0 + tz^1$, which adds noise to the ground-truth sample, by sampling a reward-to-go $z^1 \sim R(\cdot \mid x, a)$ and a noise sample $z^0 \sim \mathcal{N}(0, I_d)$. The reward-to-go sample $z^1$ can be sampled from the dataset or a target version of the learned reward model $R_{\bar{\theta}}(\cdot \mid x, a)$. We sample a reward-to-go from $R_\theta(\cdot \mid x, a)$ using the standard forward Euler method applied on the learned flow model $s_\theta(\cdot \mid x, a, t)$.

# B  FULL RESULTS

Table 2: **EVOR**'s **Overall Performance By Task.** We present the full results on each OGBench task. (⋆) indicates the default task in each environment. Results are averaged over five seeds per task for OGBench and three seeds per task for D4RL, with standard deviations reported. `IQL`'s D4RL results are reported from prior work Tarasov et al. (2023b).

| Task | IQL | QC-1 | QC-5 | EVOR |
|---|---|---|---|---|
| OGBench antmaze-large-navigate-singletask-task1-v0 (⋆) | **56** $_{\pm 11}$ | 3 $_{\pm 3}$ | 2 $_{\pm 1}$ | 22 $_{\pm 11}$ |
| OGBench antmaze-large-navigate-singletask-task2-v0 | 21 $_{\pm 13}$ | 0 $_{\pm 0}$ | 0 $_{\pm 0}$ | **62** $_{\pm 5}$ |
| OGBench antmaze-large-navigate-singletask-task3-v0 | **73** $_{\pm 4}$ | 43 $_{\pm 26}$ | 28 $_{\pm 11}$ | 32 $_{\pm 4}$ |
| OGBench antmaze-large-navigate-singletask-task4-v0 | 19 $_{\pm 11}$ | 0 $_{\pm 0}$ | 0 $_{\pm 0}$ | **65** $_{\pm 9}$ |
| OGBench antmaze-large-navigate-singletask-task5-v0 | 56 $_{\pm 5}$ | 0 $_{\pm 0}$ | 4 $_{\pm 3}$ | **69** $_{\pm 5}$ |
| OGBench antmaze-large-stitch-singletask-task1-v0 (⋆) | **19** $_{\pm 13}$ | 3 $_{\pm 2}$ | 3 $_{\pm 2}$ | 0 $_{\pm 0}$ |
| OGBench antmaze-large-stitch-singletask-task2-v0 | **1** $_{\pm 1}$ | 0 $_{\pm 0}$ | 0 $_{\pm 0}$ | **1** $_{\pm 0}$ |
| OGBench antmaze-large-stitch-singletask-task3-v0 | **70** $_{\pm 12}$ | 28 $_{\pm 31}$ | 21 $_{\pm 16}$ | 67 $_{\pm 5}$ |
| OGBench antmaze-large-stitch-singletask-task4-v0 | 0 $_{\pm 0}$ | 0 $_{\pm 0}$ | 0 $_{\pm 0}$ | **5** $_{\pm 1}$ |
| OGBench antmaze-large-stitch-singletask-task5-v0 | **31** $_{\pm 8}$ | 15 $_{\pm 14}$ | 16 $_{\pm 5}$ | 4 $_{\pm 3}$ |
| OGBench cube-double-play-100M-singletask-task1-v0 (⋆) | 0 $_{\pm 0}$ | 88 $_{\pm 6}$ | 78 $_{\pm 11}$ | **96** $_{\pm 2}$ |
| OGBench cube-double-play-100M-singletask-task2-v0 | 88 $_{\pm 7}$ | 55 $_{\pm 4}$ | 54 $_{\pm 24}$ | **93** $_{\pm 4}$ |
| OGBench cube-double-play-100M-singletask-task3-v0 | 83 $_{\pm 11}$ | 49 $_{\pm 11}$ | 56 $_{\pm 26}$ | **94** $_{\pm 6}$ |
| OGBench cube-double-play-100M-singletask-task4-v0 | 6 $_{\pm 3}$ | 23 $_{\pm 4}$ | **38** $_{\pm 17}$ | 32 $_{\pm 8}$ |
| OGBench cube-double-play-100M-singletask-task5-v0 | 43 $_{\pm 9}$ | 60 $_{\pm 7}$ | 59 $_{\pm 26}$ | **90** $_{\pm 6}$ |
| OGBench pointmaze-medium-navigate-singletask-task1-v0 (⋆) | **98** $_{\pm 2}$ | 97 $_{\pm 3}$ | **99** $_{\pm 1}$ | **99** $_{\pm 1}$ |
| OGBench pointmaze-medium-navigate-singletask-task2-v0 | 0 $_{\pm 0}$ | 85 $_{\pm 15}$ | **100** $_{\pm 1}$ | 99 $_{\pm 2}$ |
| OGBench pointmaze-medium-navigate-singletask-task3-v0 | **100** $_{\pm 1}$ | 98 $_{\pm 3}$ | 99 $_{\pm 2}$ | 99 $_{\pm 2}$ |
| OGBench pointmaze-medium-navigate-singletask-task4-v0 | **100** $_{\pm 0}$ | 76 $_{\pm 39}$ | **100** $_{\pm 0}$ | **100** $_{\pm 0}$ |
| OGBench pointmaze-medium-navigate-singletask-task5-v0 | **100** $_{\pm 0}$ | **100** $_{\pm 0}$ | **100** $_{\pm 0}$ | **100** $_{\pm 0}$ |
| OGBench scene-play-sparse-singletask-task1-v0 | 85 $_{\pm 2}$ | 93 $_{\pm 2}$ | **100** $_{\pm 0}$ | **100** $_{\pm 0}$ |
| OGBench scene-play-sparse-singletask-task2-v0 (⋆) | 46 $_{\pm 14}$ | 85 $_{\pm 5}$ | **99** $_{\pm 1}$ | 98 $_{\pm 1}$ |
| OGBench scene-play-sparse-singletask-task3-v0 | 19 $_{\pm 11}$ | 52 $_{\pm 15}$ | **93** $_{\pm 3}$ | 89 $_{\pm 5}$ |
| OGBench scene-play-sparse-singletask-task4-v0 | 40 $_{\pm 10}$ | 4 $_{\pm 4}$ | **91** $_{\pm 3}$ | 84 $_{\pm 24}$ |
| OGBench scene-play-sparse-singletask-task5-v0 | 0 $_{\pm 0}$ | 0 $_{\pm 0}$ | 33 $_{\pm 16}$ | **64** $_{\pm 17}$ |
| D4RL antmaze-large-diverse-v2 | 30 | **40** $_{\pm 28}$ | 20 $_{\pm 4}$ | 37 $_{\pm 7}$ |
| D4RL antmaze-large-play-v2 | **42** | 31 $_{\pm 16}$ | 21 $_{\pm 13}$ | 39 $_{\pm 17}$ |

## Q: What is **EVOR**'s task-level performance?

Across six environments and 27 unique tasks, EVOR achieves the best overall performance compared to the baselines. EVOR's expressive value learning method outperforms standard value learning methods. From the results in Table 2, we observe that EVOR outperforms or matches standard value function learning methods (QC), even compared to a method that employs action chunking (QC-5), suggesting that expressive value learning can improve performance over standard value function learning.

# C ABLATION STUDIES

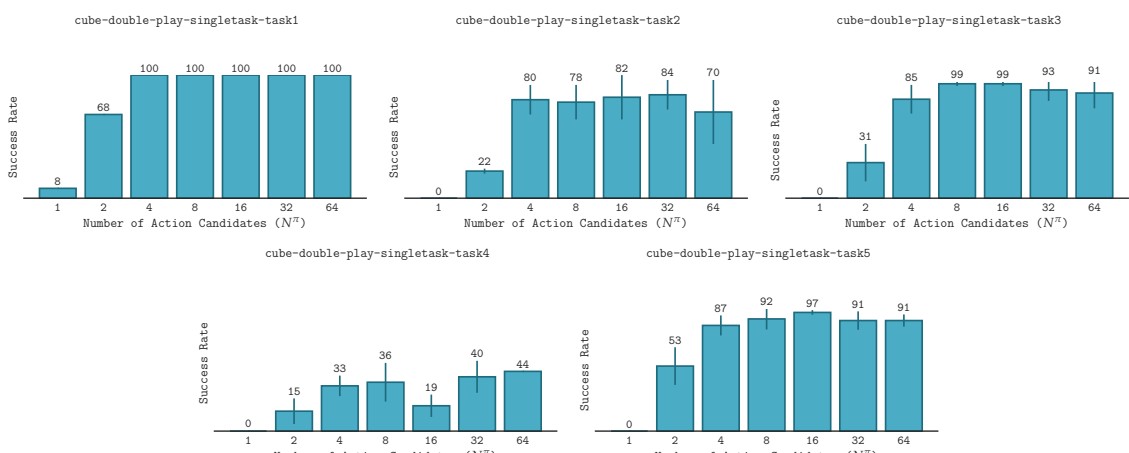

Figure 3: **Ablation Over Number of Action Candidates** $N_\pi$. Results are averaged over three seeds per task, with standard deviations reported.

**Q: [Task-Level] How can EVOR take advantage of greater inference-time compute?**

As shown in Figure 3, when given access to greater inference-time compute, EVOR can increase the number of action candidates $N_\pi$, resulting in better performance (up to a saturation point).

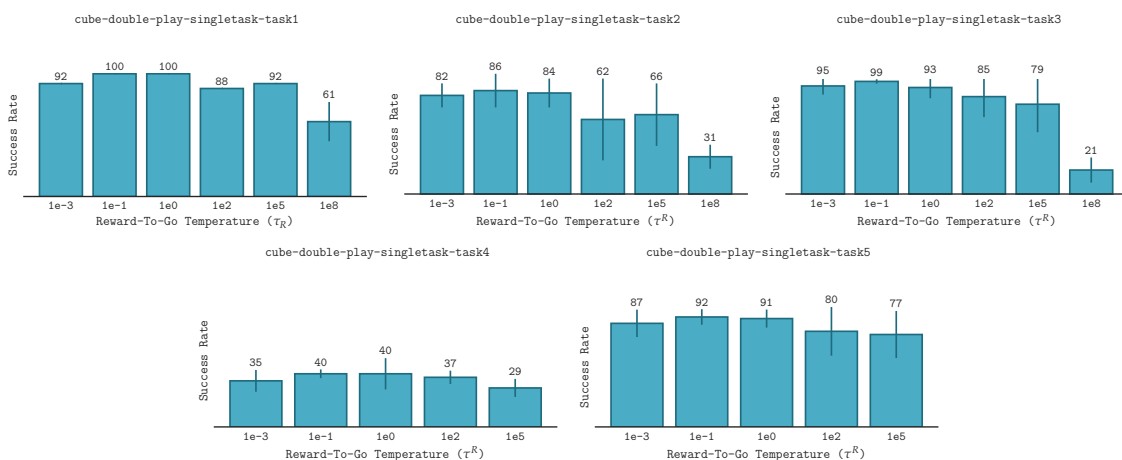

Figure 4: **Ablation Over Reward-To-Go Temperature Parameter** $\tau_R$. Results are averaged over three seeds per task, with standard deviations reported.

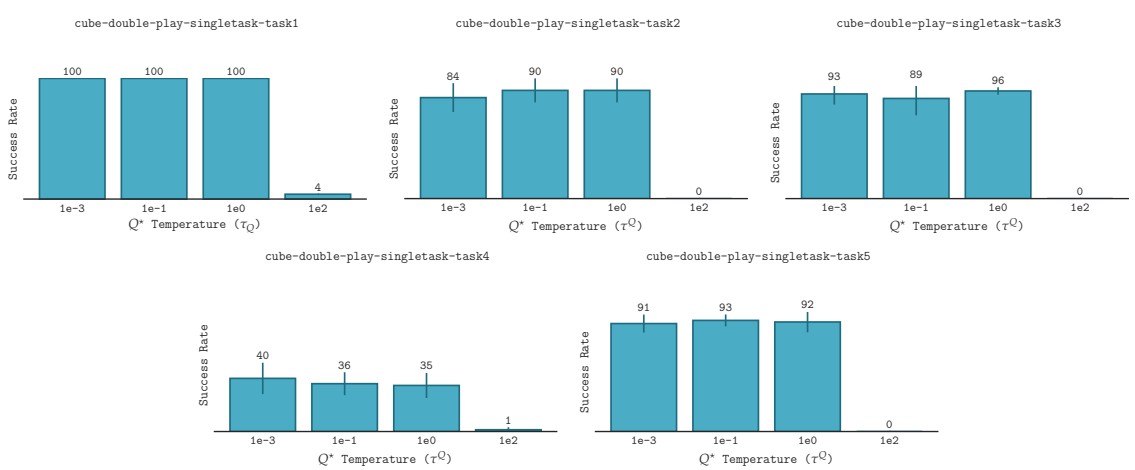

Figure 5: **Ablation Over** $Q^\star$ **Temperature Parameter** $\tau_Q$. Results are averaged over three seeds per task, with standard deviations reported.

**Q: [Task-Level] How can EVOR perform inference-time regularization?**

As shown in Figure 4 and Figure 5, by increasing varying the temperature parameters $\tau_R$ and $\tau_Q$, EVOR can vary the level of regularization to the base policy compared to the level of policy optimization. As $\tau_Q$ decreases, the action selection becomes more greedy, while as $\tau_Q$ increases, the action selection becomes more regularized. Set to a high value, EVOR becomes equivalent to the base policy (i.e., the performance with $N_\pi = 1$).

# D  EXPERIMENTAL AND IMPLEMENTATION DETAILS

In this section, we describe the setup, implementation details, and baselines used in the paper.

## D.1  EXPERIMENTAL SETUP

We follow OGBench's official evaluation scheme (Park et al., 2024a), with the reward-maximizing offline setup of Park et al. (2025b); Espinosa-Dice et al. (2025). We restate the experimental setup here. Following Park et al. (2025b); Espinosa-Dice et al. (2025), we use OGBench's `singletask` variants for all experiments, corresponding to reward-based tasks that are suitable for our reward-maximizing offline RL setting.

**Environments and Tasks.**  EVOR is evaluated on manipulation and locomotion robotics tasks in version `1.1.0` of OGBench (Park et al., 2024a), including

1. `antmaze-large-navigate-singletask-v0`
2. `antmaze-large-stitch-singletask-v0`
3. `cube-double-play-singletask-v0`
4. `pointmaze-medium-navigate-singletask-v0`
5. `scene-play-singletask-v0`

We use the three unique tasks (e.g., `antmaze-large-navigate-singletask-task{1,2,3,4,5}-v0`) for each environment listed above, where each task provides a unique evaluation goal. Each environment's dataset is labeled with a semi-sparse reward (Park et al., 2024a), and we use the `sparse` reward for `scene-play`, following Li et al. (2025). For the `cube-double-play` environment, we use the 100M size dataset provided by Park et al. (2025a).

The selected environments consist of locomotion and manipulation control problems. The `antmaze` tasks consist of navigating a quadrupedal agent (8 degrees of freedom) through complex mazes. The `cube` and `scene` environments manipulated objects with a robotic arm. The goal of `scene` tasks is to sequence multiple subtasks. The environments are state-based. We test both `navigate` and `stitch` datasets for locomotion and `play` for manipulation. These datasets are built from suboptimal, goal-agnostic trajectories, which poses a challenge for goal-directed policy learning (Park et al., 2024a). Following Park et al. (2025b); Espinosa-Dice et al. (2025), we evaluate agents using binary task success rates (i.e., goal completion percentage), which is consistent with OGBench's evaluation setup (Park et al., 2024a).

**Evaluation.**  We follow OGBench's official evaluation scheme (Park et al., 2024a). Algorithms are trained for 1,000,000 gradient steps and evaluated on 50 episodes every 100,000 gradient steps. The average success rates of the final three evaluations (i.e., the evaluation results at 800,000, 900,000, and 1,000,000 gradient steps) are reported. Tables average over 3 seeds per task and report standard deviations, bolding values within 95% of the best performance.

## D.2  EVOR IMPLEMENTATION DETAILS

**Flow Matching.**  EVOR is implemented on top of Li et al. (2025)'s open-source implementation of QC, which is adapted from Park et al. (2024a)'s open-source codebase. We implement flow matching using the same, standard velocity field as QC.

**Network Architecture and Optimizer.**  Following Park et al. (2025b); Espinosa-Dice et al. (2025); Li et al. (2025), we use a multi-layer perceptron with 4 hidden layers of size 512 for both the value and policy networks. We apply layer normalization (Ba et al., 2016) to value networks and use the Adam optimizer (Kingma, 2014). All of these parameters are shared between EVOR and the baselines.

**Hyperparameters.**  We use the same hyperparameters for both EVOR and QC. Unlike many offline RL algorithms (Park et al., 2025b; Espinosa-Dice et al., 2025), EVOR does not change training parameters between environments in this paper. EVOR uses $N = 1$ during training (instead of the $N > 1$ used during evaluation) for better efficiency.

---

**Algorithm 3:** $\pi_{\text{base}}$ Action Sampling via Forward Euler Method

---

**Input:** State $x$, number of inference steps $M$
**Output:** Action $a$
$a \sim \mathcal{N}(0, I)$                                                     `Sample starting action noise`
$t \leftarrow 0$
**for** $m \in \{0, \dots, M\}$ **do**
    $a \leftarrow a + \frac{1}{M} v_\phi(a, t, \mid x)$                                    `Follow ODE`
    $t \leftarrow t + \frac{1}{M}$
**return** $a$

---

**Inference Procedure.** EVOR's inference procedure is shown in Algorithm 2. Actions are sampled from the base policy $\pi_{\text{base}}$ via the forward Euler method, shown in Algorithm 3.

Recall that the *optimal* $Q$-function is given by:

$$Q^\star(x, a) = \eta \ln \mathbb{E}_{r \sim R(\cdot \mid x, a)} \exp(r/\eta), \tag{37}$$

where $R$ is the conditional distribution of rewards-to-go under $\pi_{\text{ref}}$. We learn an estimate of $R$ via the flow-based TD objective, such that $R_\theta(\cdot \mid x, a) \approx R(\cdot \mid x, a)$. We approximate the expectation via sample averaging, as shown in Algorithm 2, such that

$$\text{LogSumExp}(z^{(j)}) = \tau^\star \log \frac{1}{N} \sum_{j=1}^{N} \exp\left(\frac{z^{(j)}}{\tau^\star}\right) \tag{38}$$

We then construct a weighted softmax via the $Q^\star$ approximation, such that

$$\text{softmax}(Q^\star(x, a^{(j)})) = \frac{\exp(Q^\star(x, a^{(j)})/\tau)}{\sum_{j=1}^{N_\pi} \exp(Q^\star(x, a^{(j)})/\tau)} \tag{39}$$

Table 3: Shared Hyperparameters Between QC Baselines and EVOR.

| PARAMETER | VALUE |
|---|---|
| LEARNING RATE | 3E-4 |
| OPTIMIZER | ADAM (KINGMA, 2014) |
| GRADIENT STEPS | 1E6 |
| MINIBATCH SIZE | 256 |
| MLP DIMENSIONS | [512, 512, 512, 512] |
| TARGET NETWORK SMOOTHING COEFFICIENT | 5E-3 |
| DISCOUNT FACTOR $\gamma$ | 0.99 |
| DISCRETIZATION STEPS | 10 |
| TIME SAMPLING DISTRIBUTION | UNIF([0,1]) |
| NUMBER OF ACTION CANDIDATES $N_\pi$ | 32 |

Table 4: Hyperparameters for EVOR.

| HYPERPARAMETER | VALUE |
|---|---|
| BETA $\beta$ | 1E-3 |
| $Q^\star$ BETA $\beta^\star$ | 1 |
| NUMBER OF RTG SAMPLES $N^{\mathrm{RTG}}$ | 1 (TRAIN), 50 (EVAL) |

Table 5: Hyperparameters for QC.

| HYPERPARAMETER | VALUE |
|---|---|
| ACTION CHUNK LENGTH | 1 (QC-1), 5 (QC-5) |
| CRITIC ENSEMBLE SIZE | 2 |

## D.3 BASELINES

Rather than compare to all of the existing offline RL algorithms benchmarked on OGBench, we instead aim to isolate the effect of expressive value learning in order to demonstrate its benefit specifically.

**Q-Chunking (QC).** We compare to $Q$-chunking (QC, Li et al. (2025)), a recent offline RL algorithm that is closest to EVOR. Like EVOR, QC learns a base policy via flow matching and extracts an optimized policy via rejection sampling. The key difference between QC and EVOR is in how the value function is learned, which is the exact difference we aim to isolate. QC can employ action chunking in both its policy and value function, and we compare EVOR to both QC with (QC-5) action chunking and without it (QC-1). We select the action chunk length (5) based on Li et al. (2025)'s recommendation.

Given an action chunk length of $k$, represented as $\boldsymbol{a}_{t:t+k} = (a_t, a_{t+1}, \ldots, a_{t+k})$, the $Q$-function is updated via

$$Q(x_t, \boldsymbol{a}_{t:t+k}) \leftarrow \sum_{t'=1}^{t+k-1} [r_{t'}] + Q(x_{t+k}, \boldsymbol{a}_{t+k:t+2k}) \tag{40}$$

and actions are sampled via

$$\boldsymbol{a} \leftarrow \operatorname*{argmax}_{\boldsymbol{a} \in \{\boldsymbol{a}_1, \boldsymbol{a}_2, \ldots, \boldsymbol{a}_N\}} Q(x, \boldsymbol{a}), \tag{41}$$

where $a_1, a_2, \ldots, a_N \sim \pi_{\text{base}}(\cdot \mid x)$. This yields the following loss function for learning the $Q$-function:

$$L(\theta) = \mathbb{E}_{\substack{x_t, \boldsymbol{a}_t \sim \mathcal{D} \\ \{a_{t+k}^i\}_{i=1}^N \sim \pi_{\text{base}}(\cdot \mid x_{t+k})}} \left[ \left( Q_\theta(x_t, \boldsymbol{a}_t) - \sum_{t'=1}^{k} r_{t+t'} - Q_{\bar{\theta}}(x_{t+k}, \boldsymbol{a}_{t+k}) \right)^2 \right], \tag{42}$$

where $\boldsymbol{a}_{t+k} = \operatorname{argmax}_{a \in \{\boldsymbol{a}_{t+k}^i\}} Q(s, \boldsymbol{a})$.

**Implicit Q-Learning (IQL).** Additionally, we add Implicit Q-Learning (IQL) (Kostrikov et al., 2022) as an additional baseline, which presents a separate approach for learning a value function, specifically through expectile regression. IQL then extracts the policy through advantage-weighted regression (Peters & Schaal, 2007; Wang et al., 2018; Peng et al., 2019; Nair et al., 2020). For the $\alpha$ parameter in IQL, we use the values from Kostrikov et al. (2022); Park et al. (2025b).

