# OpenReview forum: "Expressive Value Learning for Scalable Offline Reinforcement Learning"
_ICLR.cc/2026/Conference — Submitted to ICLR 2026_

### Official Review · Reviewer_sTEA · 2025-10-25

**Soundness:** 2
**Presentation:** 2
**Contribution:** 2
**Rating:** 4
**Confidence:** 3

**Summary:**

This work presents a generative offline RL algorithm, which avoids policy distillation and backpropagation through time during policy optimization. Experiments on OGBench validate the effectiveness of the proposed method.

**Strengths:**

- EVOR enables inference-time policy extraction without relying on model distillation or backpropagation through time.
- The proposed approach outperforms baselines in various benchmarks.
- This paper provides both empirical and theoretical analyses.

**Weaknesses:**

- The authors argue that one-step models are difficult to scale to larger base policies (e.g., VLAs) or real-world tasks. However, no corresponding experiments on VLA or real-world settings are provided to support this claim.
- The paper includes only a single baseline (Q-chunking) and lacks comparisons with other competitive methods, such as FQL [1].
- The evaluation is limited to OGBench. It would be convincing to include results on other benchmarks, such as D4RL [2].

References:

[1] Park et al. "Flow Q-Learning", ICML, 2025.

[2] Fu et al. "D4RL: Datasets for Deep Data-Driven Reinforcement Learning", arXiv preprint arXiv:2004.07219, 2020.

**Questions:**

Have you compared the training time and the number of parameters with non-generative offline methods such as CQL [1]?

Reference:

[1] Kumar et al. "Conservative Q-Learning for Ofﬂine Reinforcement Learning", NeurIPS, 2020.

---

> ### Author Response · Authors · 2025-11-25
> **Response to**
>
> We thank the reviewer for their feedback. We address their comments and questions below.
>
> ## 1. One-Step Models
>
> > The authors argue that one-step models are difficult to scale to larger base policies (e.g., VLAs) or real-world tasks...
>
> In the computer vision domain, **[1] argues that two-stage approaches**, where a larger model is distilled into a one-step student model, and **consistency-based approaches require bootstrapping and carefully constructed learning schedules**, and they are **less expressive than multi-step approaches**.
>
> In the robotics simulation, **[3] shows that leveraging multi-step models outperforms one-step distillation-based methods** like FQL and CAC.
>
> In the real-world robotics and VLA domain, **[4] argues for a rejection sampling-based approach to optimizing VLAs**.
>
> ## 2. Baselines
> > The paper includes only a single baseline (Q-chunking)...
>
> > Have you compared the training time and the number of parameters with non-generative offline methods such as CQL [1]?
>
> We wish to emphasize **the goal of our experimental section: we aim to focus on value learning, specifically investigating whether our method of expressive value learning improves over standard value learning methods**. Comparing to methods like FQL would introduce an orthogonal axis – a separate policy extraction method – that would confound our evaluation of the value learning method. Put simply, **we want our baselines to use the same policy extraction method so that we can target the differences in value learning**.
>
> Moreover, we explicitly state that our aim is not to evaluate the policy extraction methods, and we do not claim that our rejection sampling-based method of policy extraction is the best. We further state that integrating our value learning method into other forms of policy extraction is left for future work.
>
> However, to help address the reviewer’s concerns, **we added IQL [5] as a baseline** because it presents a different method of value learning, specifically one based on expectile regression. We note that IQL outperformed CQL on the selected environments [5].
>
> ## 3. Evaluation
> > The evaluation is limited to OGBench. It would be convincing to include results on other benchmarks, such as D4RL [2]
>
> **We added D4RL tasks [2] to the evaluation**, and **our method achieves the best overall performance on this benchmark** as well.
>
> ## Citations
>
> [1] Frans et al. “One Step Diffusion via Shortcut Models”, 2024.
>
> [2] Fu et al. “D4RL: Datasets for Deep Data-Driven Reinforcement Learning”, 2020.
>
> [3] Espinosa-Dice et al. “Scaling Offline RL via Efficient and Expressive Shortcut Models”, 2025.
>
> [4] Nakamoto et al. “Steering Your Generalists: Improving Robotic Foundation Models via Value Guidance”, 2025.
>
> [5] Kostrikov et al. "Offline Reinforcement Learning with Implicit Q-Learning", 2021

---

> > ### Comment · Reviewer_sTEA · 2025-11-26
> >
> > Thanks for your responses. However, this work still lacks a strong FQL baseline and does not compare training time as well as the number of parameters with non-generative offline methods such as IQL. Therefore, I will keep my rating.

---

> > > ### Author Response · Authors · 2025-12-04
> > >
> > > Thank you for your response. We respond to the remaining comments below.
> > >
> > > > ... this work still lacks a strong FQL baseline
> > >
> > > We are **focused on determining whether our method of expressive value learning is beneficial over standard value learning methods**. FQL uses a different policy extraction method, which is not what our experimental section wishes to investigate. Put simply, we want our baselines to use the same policy extraction method so that we can target the differences in value learning. **We even suggest an avenue for integrating our value learning method into other forms of policy extraction**: recent work [4] has integrated policy gradients and flow-based policies by first distilling the expressive, multi-step model into a one-step model. Then, this lightweight one-step model is used to generate actions to optimize the Q function in Q learning. We could leverage a similar approach, where we distill the expressive, multi-step model of our base policy into a one-step model. Then, we could perform Q learning with our learned, optimal Q function in a similar style.
> > >
> > > > ... does not compare training time as well as the number of parameters with non-generative offline methods such as IQL
> > >
> > > **The size of the policy and value networks are the same between EVOR and the baselines**. We will include a comparison of runtime in the final version of the paper.

---

### Official Review · Reviewer_EQof · 2025-10-28

**Soundness:** 3
**Presentation:** 2
**Contribution:** 3
**Rating:** 6
**Confidence:** 2

**Summary:**

This paper proposes EVOR (Expressive Value Learning for Offline Reinforcement Learning), a scalable offline RL framework that integrates expressive generative models (flow matching) into both the policy and the value function.
The authors identify a key bottleneck in recent diffusion- and flow-based offline RL: Existing methods either rely on backpropagation through time (BPTT), which is computationally expensive, or on policy distillation, which introduces compounding errors and limits scalability to larger base policies (e.g., vision-language-action models).
EVOR addresses this by: 1. Learning expressive value functions via flow-based temporal difference (TD) learning, avoiding Gaussian approximations 2. Performing inference-time policy extraction through rejection sampling against the expressive learned value function, enabling scalable optimization and regularization without retraining. 3. Leveraging inference-time compute scaling  and regularization. EVOR thus provides a unified framework for scalable, expressive, and compute-efficient offline RL, avoiding BPTT and distillation entirely. Empirically, EVOR significantly outperforms the baseline Q-Chinking (QC) across 25 tasks in the OGBench suite (antmaze, cube manipulation, scene play, and pointmaze). It also demonstrates strong scaling properties with increased inference compute and robustness to hyperparameters.

**Strengths:**

The paper addresses a timely and underexplored issue: how to scale offline RL with expressive generative models without incurring the computational and stability issues of BPTT or distillation. While diffusion and flow models have been extensively used for policy modeling, they have not been effectively integrated into value learning. EVOR’s insight, that flow-based value functions can provide a scalable and expressive mechanism for both regularization and inference-time optimization, is conceptually elegant and fills an important gap.
The theoretical formulation is solid and well-grounded, and this clarity and generality make EVOR theoretically appealing and reproducible.
Unlike prior work that focuses on expressive policies, EVOR emphasizes expressive value functions, a neglected component in scaling offline RL. This focus on value expressivity represents a genuine conceptual advance and could influence future RL architectures.
EVOR achieves consistent improvements across many tasks, outperforming QC-1 and QC-5 baselines. Importantly, EVOR uses the same training hyperparameters across all environments.

**Weaknesses:**

Although the authors justify focusing on QC to isolate “value learning expressivity,” it would be more compelling to include additional baselines such as IQL and CQL.
All results are on simulation-based OGBench tasks. Given the paper’s claim of “scalability,” evaluation on higher-dimensional or real-robot tasks (e.g., Pi0, RT-1 datasets) would strengthen the impact. Even a discussion on computational resource scaling (training time, memory vs. QC) would help quantify efficiency.
While the ablations on inference-time parameters are thorough, there is no study isolating the contribution of flow-based TD learning itself versus standard TD learning with a Gaussian critic. A simple comparison of “EVOR without flow-based TD” would help clarify whether expressivity, or the training dynamics, drive the observed improvements.
The derivation of the optimal regularized Q-function (Eq. 16) assumes deterministic transitions, which rarely hold in realistic robotics. Although the authors later extend to stochastic settings (Section 3.2), a discussion of approximation error or theoretical relaxation would improve rigor.

**Questions:**

1.	It would be better if the authors add ablations on flow-based TD learning, compare to Gaussian critics or classification-based Q-updates (Farebrother et al., 2024) to show explicit benefit.
2.	It would be better if the authors include wall-clock efficiency metrics to show training/inference time versus QC and policy distillation methods.
3.	It would be better if the authors discuss potential integration with policy gradients.  The discussion section (p. 9) hints at combining EVOR with reparameterized gradients; elaborating on this could position EVOR as a foundation for hybrid methods.

---

> ### Author Response · Authors · 2025-11-25
> **Response to Reviewer EQof**
>
> We thank the reviewer for their feedback. We address their comments and questions below.
>
> ## 1. Baselines
> > Although the authors justify focusing on QC to isolate “value learning expressivity,” it would be more compelling to include additional baselines such as IQL and CQL.
>
> To address the reviewer’s concerns, **we added IQL [1] as a baseline**, and we note that IQL was shown to outperform CQL on the selected environments [2]. The results show that **EVOR outperforms IQL across the six environments and 27 tasks** considered.
>
> ## 2. Environments
> > All results are on simulation-based OGBench tasks.
>
> To address the reviewer's concern, **we added D4RL tasks [3] to the evaluation**, and we show that **EVOR achieves the best overall performance on this benchmark** as well.
>
> > Given the paper’s claim of “scalability,” evaluation on higher-dimensional or real-robot tasks... would strengthen the impact...
>
> We agree with the reviewer that there are additional axes along which we can demonstrate scalability, including real-world robot tasks with frontier VLA models. However, **real-world robot experiments are out of the scope of this paper**, as our experimental section is intending to target the benefit of expressive value learning over standard TD learning methods, specifically in simulation. Real-world experiments would introduce additional confounding factors that would affect such a comparison, and we agree that this presents interesting avenues for future work.
>
> ## 3. Flow-Based TD vs Standard TD Learning
> > While the ablations on inference-time parameters are thorough, there is no study isolating the contribution of flow-based TD learning itself versus standard TD learning with a Gaussian critic...
>
> > It would be better if the authors add ablations on flow-based TD learning... to show explicit benefit.
>
> We wish to clarify that **we do isolate the contribution of flow-based TD learning in our experimental section**. Specifically, the baseline we choose (QC) has uses the same policy extraction method (rejection sampling against the Q function) but with a different value learning method (standard TD). Our method requires sample averaging the expectation, but as detailed in Appendix D and Table 4, we set the temperature parameter so that the policy chooses the greedy action. Thus, we believe **our baseline satisfies the reviewer's request for a comparison of flow-based TD learning with standard TD learning**.
>
> Additionally, to address the reviewer’s concerns, we added IQL [1] as a baseline because it presents a different method of value learning based on expectile regression.
>
> ## 4. Derivation of Optimal $Q$-Function
> > The derivation of the optimal regularized Q-function (Eq. 16) assumes deterministic transitions, which rarely hold in realistic robotics. Although the authors later extend to stochastic settings (Section 3.2), a discussion of approximation error or theoretical relaxation would improve rigor.
>
> The assumption of deterministic dynamics is frequently used in the analysis of offline RL algorithms (Edwards et al., 2020; Ma
> et al., 2022; Schweighofer et al., 2022; Park et al., 2023; Ghosh et al., 2023; Wang et al., 2023; Karabag & Topcu, 2023;
> Park et al., 2024a;c). However, because our method leverages (a flow-based variant of) TD learning, **it can work in the stochastic setting**.
>
> ## Q. Efficiency Metrics
> > It would be better if the authors include wall-clock efficiency metrics to show training/inference time versus QC and policy distillation methods.
>
> In this paper, we explicitly aim to avoid policy distillation-based approaches. Their main issue is not that the wall-clock efficiency but that simple, lightweight one-step distilled models may struggle as the base model is scaled up.
>
> ## Q. Discussion of Integration with Policy Gradients
> > It would be better if the authors discuss potential integration with policy gradients. The discussion section (p. 9) hints at combining EVOR with reparameterized gradients; elaborating on this could position EVOR as a foundation for hybrid methods.
>
> Recent work [4] has integrated policy gradients and flow-based policies by first distilling the expressive, multi-step model into a one-step model. Then, this lightweight one-step model is used to generate actions to optimize the Q function in Q learning. We could leverage a similar approach, where we distill the expressive, multi-step model of our base policy into a one-step model. Then, we could perform Q learning with our learned, optimal Q function in a similar style.
>
> ## Citations
>
> [1] Kostrikov et al. "Offline Reinforcement Learning with Implicit Q-Learning", 2021
>
> [2] Park et al. “OGBench: Benchmarking Offline Goal-Conditioned RL”, 2024.
>
> [3] Fu et al. “D4RL: Datasets for Deep Data-Driven Reinforcement Learning”, 2020.
>
> [4] Park et al. “Flow Q-Learning”, 2025.

---

### Official Review · Reviewer_PRpU · 2025-10-31

**Soundness:** 3
**Presentation:** 4
**Contribution:** 3
**Rating:** 6
**Confidence:** 3

**Summary:**

This paper proposes EVOR, a novel flow-matching-based offline reinforcement learning algorithm that scales to data, model, and inference-time computation without enforcing the backpropagation through time or policy distillation. EVOR introduces expressive policy and value learning via flow-based temporal difference learning with the distributional Bellman operator and the reward-to-go approach. Furthermore, EVOR enables inference-time-scalable offline RL by leveraging rejection sampling with the learned optimal value function and searching the test-time hyperparameters (e.g., $\tau$ temperature) without further training. Experiments highlight the contribution of expressive value learning against the baseline with flow matching policy learning in OGBench.

**Strengths:**

- The paper is well structured and sounds coherent.
- EVOR demonstrates strong performance on diverse tasks in OGBench.
- The authors provide empirical observations on EVOR's inference-time scaling property by carefully designing ablation studies.

**Weaknesses:**

- The novelty of EVOR is marginal. The theoretical analysis of expressive flow matching policy learning strictly depends on Theorem 2.2 in Zhou et al. While it is noticeable that EVOR suggests a novel expressive value learning by introducing a flow-based TD objective with reward-to-go, experimental results on verifying the performance of EVOR on diverse offline RL tasks are insufficient to claim the practicality of EVOR. I appreciate the main results on comparing QC with EVOR and the additional ablation studies on inference-time scaling, while I believe further experiments on differentiating EVOR with state-of-the-art offline RL methods (specifically, offline RL with generative models) would support the novelty of this paper.
- It is unclear how EVOR scales to different axes of scalability: the data and model in offline RL. The authors introduce three heterogeneous axes for scaling offline RL in the introduction: the data, model, and computing. However, the primary results in the paper mainly consider the inference-time scaling property. Offline trajectories often involve multi-modal, sub-optimal behaviors from unknown policies, which makes it hard for the model to learn such diverse behaviors with limited scalability. While OGBench provides diverse tasks with thoroughly collected sub-optimal behaviors, comparing only the performance of EVOR in single tasks may overshadow the true scalability of the data and model.

**Questions:**

- Could you provide additional results with popular baselines in OGBench? As the authors denote in the paper, comparing with baselines trained with standard offline RL objectives may escape the scope of this paper. However, comparing with diffusion-based (or flow-based) policy learning [1, 2] (representatives for BPTT) or policy distillation methods [3] (representatives for distillation) would provide an in-depth view of scalable offline RL with generative models.
- Could you visualize how EVOR learns complex behaviors from the dataset using flow matching TD objectives and policy extractions? For instance, Diffusion Policy [4] demonstrates strong multi-modal capability by visualizing modes in trajectory levels.

# Minor problems (do not affect the score)
- There are several typos:
- - In L86, does not learn require $\rightarrow$ does not require
- - In L91, the Q-function learned $\rightarrow$ the learned Q-function
- - In 199, samples from in $\rightarrow$ samples from
- The Related Work section is duplicated in the Appendix. I appreciate the extended related work section in the Appendix, but offline RL with generative models and inference-time scaling in offline RL are exactly the same as the section in the main paper.

[1] Park S, Li Q, Levine S. Flow q-learning. arXiv 2025.

[2] Hansen-Estruch P, Kostrikov I, Janner M, Kuba JG, Levine S. Idql: Implicit q-learning as an actor-critic method with diffusion policies. arXiv 2023.

[3] Ding Z, Jin C. Consistency models as a rich and efficient policy class for reinforcement learning. arXiv 2023.

[4] Chi C, Xu Z, Feng S, Cousineau E, Du Y, Burchfiel B, Tedrake R, Song S. Diffusion policy: Visuomotor policy learning via action diffusion. The International Journal of Robotics Research 2025.

---

> ### Author Response · Authors · 2025-11-25
> **Response to Reviewer PRpU**
>
> We thank the reviewer for their feedback. We address their comments and questions below.
>
> ## 1. Novelty
> > The novelty of EVOR is marginal...
>
> We emphasize that **[1] is focused on the language setting with deterministic dynamics and assumes access to environment interactions**. Our paper, in contrast, focuses on the **offline RL and robotics settings**. We **introduce a novel method of expressive value learning**, which was not present in [1], and **it can work in the stochastic setting by building on TD learning**. Therefore, we believe this contribution is both useful and unique from prior work.
>
> ## 2. Comparisons and Baselines
> > While it is noticeable that EVOR suggests a novel expressive value learning by introducing a flow-based TD objective... experimental results on verifying the performance of EVOR on diverse offline RL tasks are insufficient to claim the practicality of EVOR...
>
> > Could you provide additional results with popular baselines in OGBench?
>
> We wish to emphasize **the goal of our experimental section: we aim to focus on value learning, specifically investigating whether our method of expressive value learning improves over standard value learning methods**. Comparing to methods like FQL would introduce an orthogonal axis – a separate policy extraction method – that would confound our evaluation of the value learning method. Put simply, **we want our baselines to use the same policy extraction method so that we can target the differences in value learning**.
>
> Moreover, we explicitly state that our aim is not to evaluate the policy extraction methods, and we do not claim that our rejection sampling-based method of policy extraction is the best. We further state that integrating our value learning method into other forms of policy extraction is left for future work.
>
> However, to help address the reviewer’s concerns, **we added IQL [4] as a baseline** because it presents a different method of value learning, specifically one based on expectile regression.
>
> ## 3. Axes of Scalability
> > It is unclear how EVOR scales to different axes of scalability: the data and model in offline RL.... the primary results in the paper mainly consider the inference-time scaling property... Offline trajectories often involve multi-modal, sub-optimal behaviors from unknown policies... While OGBench provides diverse tasks with thoroughly collected sub-optimal behaviors, comparing only the performance of EVOR in single tasks may overshadow the true scalability of the data and model.
>
> The **environments we evaluate on [2, 3] offer a mix offline data quality**, and it should be noted that approaches like behavioral cloning (BC) fail to learn a competitive policy [2]. In contrast, our method learns a competitive policy that outperforms the baselines. To provide greater diversity in environments, **we added D4RL tasks [3] to the evaluation**, and **our method achieves the best overall performance on this benchmark** as well.
>
> ## 4. Typos
> > There are several typos... The Related Work section is duplicated in the Appendix...
>
> We addressed the typos. We also unified the Related Work sections into the main portion of the paper.
>
> ## Q. Visualization
> > Could you visualize how EVOR learns complex behaviors from the dataset using flow matching TD objectives and policy extractions? For instance, Diffusion Policy [6] demonstrates strong multi-modal capability by visualizing modes in trajectory levels.
>
> We emphasize that **EVOR is able to work with any base policy**, including an expressive and multi-modal diffusion base policy [6]. Because prior work [6, 7] has shown that diffusion policies have the ability to generate multi-modal action distributions, we do not feel it necessary to reproduce those results. Our method also enables greater expressivity in modeling the reward-to-go distribution, but this is a less-intuitive metric to visualize.
>
> ## Citations
> [1] Zhou et al. “Q#: Provably Optimal Distributional RL for LLM Post-Training”, 2025
>
> [2] Park et al. “OGBench: Benchmarking Offline Goal-Conditioned RL”, 2024.
>
> [3] Fu et al. “D4RL: Datasets for Deep Data-Driven Reinforcement Learning”, 2020.
>
> [4] Kostrikov et al. "Offline Reinforcement Learning with Implicit Q-Learning", 2021
>
> [5] Park et al. “Flow Q-Learning”, 2025.
>
> [6] Chi et al. “Diffusion Policy: Visuomotor Policy Learning via Action Diffusion”, 2025.
>
> [7] Ren et al. “Diffusion Policy Policy Optimization”, 2024.

---

> > ### Comment · Reviewer_PRpU · 2025-11-27
> >
> > Thank you for the response. I appreciate the extra experiments for addressing (i) an additional competitive offline RL baseline (i.e., IQL), which also considers an alternative value learning strategy, and (ii) another benchmark (D4RL; AntMaze tasks) for examining the suboptimality and diversity of the behaviors.
> >
> > Nevertheless, a few concerns remain unresolved, and I find it hard to be convinced by the authors' response. Regarding the novelty of EVOR, the authors remark that EVOR can be extended to a stochastic environment via TD learning and reward-to-go distribution, which pinpoints the difference against Q#. However, the authors have made an argument for learning optimal value functions based on the deterministic transitions in Theorem 1 (Section 4.2), and provided empirical observations of EVOR in OGBench and D4RL, where the tasks are inherently deterministic. There are no theoretical proofs or empirical results that EVOR can be effectively transferred into stochastic environments in this paper.
> >
> > In addition, I disagree with the authors that flow-matching-based policy can represent the multi-modal behavior in the offline dataset in L236 and the above response. While the authors claim that EVOR can be incorporated with any base policy (e.g., diffusion policy) for flow-based TD learning, it is unclear how well EVOR, combined with such multi-modal policies, exhibits similar multi-modality in the action-level trajectories since EVOR decouples the action selection from the base policy. Recent work (Zhai et al. 2025) in robotic manipulation shows that flow-matching policy may fail to capture the multi-modality in practice. Hence, further experiments should be conducted to support the claim in the main paper (e.g., L236 in Section 4.1).
> >
> > Overall, considering the positive improvements made during the rebuttal and remaining concerns, I will maintain the initial score.
> >
> > Zhai, Xuanran, et al. "Vfp: Variational flow-matching policy for multi-modal robot manipulation." arXiv 2025.

---

> ### Author Response · Authors · 2025-12-04
>
> Thank you for the response. We would like to add some clarifications:
>
> > Regarding the novelty of EVOR, the authors remark that EVOR can be extended to a stochastic environment via TD learning and reward-to-go distribution, which pinpoints the difference against Q#.
>
> **This is not the only difference against Q#**. We are performing **value learning with flow matching** – a novel contribution –**which Q# does not do**. Furthermore, we are **using flow-based TD learning, which Q# does not do**.
>
> > However, the authors have made an argument for learning optimal value functions based on the deterministic transitions in Theorem 1 (Section 4.2), and provided empirical observations of EVOR in OGBench and D4RL, where the tasks are inherently deterministic. There are no theoretical proofs or empirical results that EVOR can be effectively transferred into stochastic environments in this paper.
>
> It is true that we do not have theoretical proofs that EVOR under the stochastic setting. However, **the assumption of determinism is commonly applied in analysis of offline RL algorithms**. See for example: Edwards et al., 2020; Ma
> et al., 2022; Schweighofer et al., 2022; Park et al., 2023; Ghosh et al., 2023; Wang et al., 2023; Karabag & Topcu, 2023;
> Park et al., 2024a;c. Furthermore, **the OGBench and D4RL benchmarks are recent and commonly used benchmarks for offline RL algorithms**.
>
> > ...I disagree with the authors that flow-matching-based policy can represent the multi-modal behavior in the offline dataset in L236 and the above response. While the authors claim that EVOR can be incorporated with any base policy (e.g., diffusion policy) for flow-based TD learning, it is unclear how well EVOR, combined with such multi-modal policies, exhibits similar multi-modality in the action-level trajectories since EVOR decouples the action selection from the base policy. Recent work (Zhai et al. 2025) in robotic manipulation shows that flow-matching policy may fail to capture the multi-modality in practice. Hence, further experiments should be conducted to support the claim in the main paper (e.g., L236 in Section 4.1).
>
> We agree with the comment that multi-modality may be hard to learn in certain settings, but we do not make any claim otherwise. We simply emphasize two points:
> 1. **Our method is capable of working with any base policy** because the **rejection sampling policy extraction method simply requires that multiple action candidates can be sampled from the base policy**. We use flow matching in the event where a base policy is not given.
> 2. **There is theoretical [1] and empirical [2, 3] evidence that generative models like flow matching can model complex and multi-modal distributions**, better than Gaussian-based approaches. We are happy to revise the language used in relevant sentences, but we do not see this as significantly impacting the paper.
>
> [1] Block et al., "Provable Guarantees for Generative Behavior Cloning: Bridging Low-Level Stability and High-Level Behavior", 2023.
>
> [2] Ren et al. “Diffusion Policy Policy Optimization”, 2024.
>
> [3] Chi et al. “Diffusion Policy: Visuomotor Policy Learning via Action Diffusion”, 2025.

---

### Official Review · Reviewer_tSHF · 2025-11-10

**Soundness:** 2
**Presentation:** 3
**Contribution:** 2
**Rating:** 4
**Confidence:** 4

**Summary:**

This paper introduces EVOR (Expressive Value Learning for Offline Reinforcement Learning), a new scalable offline RL framework that integrates expressive generative value functions with expressive policies. Existing methods that use diffusion or flow-matching policies in offline RL often rely on backpropagation through time (BPTT) or policy distillation, both of which limit scalability.

EVOR avoids both by:
1. Learning a regularized, expressive Q-function using flow matching, resulting in an optimal solution to the KL-regularized offline RL objective.
2. Performing inference-time policy extraction using rejection sampling guided by the expressive value function.
3. Supporting inference-time scaling and regularization by adjusting the number of sampled actions and softmax temperatures.

Empirical results on OGBench (25 tasks across 5 environments) show consistent improvements over Q-Chunking (QC-1 and QC-5) baselines.

**Strengths:**

- Clear motivation and problem formulation. The authors clearly identify the scalability bottlenecks in prior generative-policy offline RL methods—specifically the reliance on BPTT or distillation. They motivate EVOR as a natural step forward: integrating expressive flow-based modeling into the value function, not just the policy (Sec. 1–3).
- Elegant inference-time optimization mechanism. EVOR’s rejection-sampling-based policy extraction (Algorithm 2, Eq. 22) allows scalable test-time optimization without retraining. The mechanism directly leverages the expressive Q function and supports compute scaling via N_\pi and temperature parameters (τ_R, τ_Q), which are empirically validated. EVOR’s inference-time scaling shows performance increasing monotonically with compute, aligning well with scaling-law intuition in modern generative RL systems. This feature enhances real-world deployability in compute-flexible settings.
- Empirical results show benefit. Table 1 and Table 2 demonstrate consistent improvement on all five OGBench environments, with particularly strong gains on difficult manipulation and navigation tasks.

**Weaknesses:**

- Limited novelty compared to prior works. The construction of the value function in EVOR essentially follows the same principle as "Q♯: Provably Optimal Distributional RL for LLM Post-training", which estimates Q^*(x,a) by sampling reward-to-go values and applying a softmax/log-sum-exp aggregation. The only difference is that EVOR learns the reward-to-go distribution via flow matching, but this idea is already very similar to TD-Flow (Farebrother et al., 2025) and Intention-Conditioned Flow Occupancy Models (ICFOM), which also learn value functions from offline data using flow-based velocity fields. Hence, the claimed contribution is incremental rather than conceptually novel.
- Restricted to KL-regularized RL formulation. EVOR is derived specifically under the KL-regularized RL objective, which is known to suffer from over-conservatism due to the KL divergence (penalizing exploration beyond the dataset distribution). This limitation may constrain the method’s ability to generalize to other regularizers (e.g., support-based ones) or to handle cases where mild extrapolation beyond the offline data is beneficial.
- Dependence on rejection sampling for policy extraction. The learned Q(s,a) is only utilized through rejection-sampling-based policy extraction, which makes the method heavily reliant on the quality of the offline dataset. When the offline data are suboptimal or sparse in high-value regions, rejection sampling cannot meaningfully exceed the dataset performance, since all sampled actions originate from the base policy. This limits EVOR’s applicability in low-quality or diverse-data regimes, where gradient-based policy improvement or explicit policy learning may be preferable.
- Comparative scope is narrow. Experiments are restricted to QC baselines and only on OGBench datasets. Other recent scalable offline RL approaches (e.g., Flow Q-Learning) are missing. Since EVOR’s main claim is “scalability,” benchmarking against state-of-the-art methods beyond QC would strengthen the empirical case.

**Questions:**

Since the paper emphasizes Scalable Offline Reinforcement Learning, could the authors further validate the effectiveness of Expressive Value Learning on the D4RL MuJoCo datasets? Compared with OGBench, the behaviors in D4RL datasets are typically more suboptimal and diverse, making them a strong testbed for scalability and robustness. Would Expressive Value Learning help improve performance under such more challenging, low-quality offline data distributions? This seems like a very meaningful way to verify the scalability claims of the proposed method.

---

> ### Author Response · Authors · 2025-11-25
> **Response to Reviewer tSHF**
>
> We thank the reviewer for their feedback. We address their comments and questions below.
>
> ## 1. Novelty
> > Limited novelty compared to prior works...
>
> We emphasize that **[1] is focused on the language setting with deterministic dynamics and assumes access to environment interactions**. Our paper, in contrast, focuses on the **offline RL and robotics settings**. We introduce a novel method of expressive value learning that was not used in Q#, and **it can work in the stochastic setting by building on TD learning**. Furthermore, our algorithm, EVOR, is unique to the TD-Flow [10] approaches. Therefore, we believe this contribution is both useful and unique from prior work.
>
> ## 2. KL-Regularized Formulation
> > Restricted to KL-regularized RL formulation...
>
> While we agree that the KL-regularized RL objective may have limitations, it remains the one of the most popular objectives in offline RL [2], robotics [3], and language modeling [4]. However, **prior works do not faithfully optimize the optimal, regularized objective**. In contrast, **EVOR does optimize the optimal, regularized objective**, improving upon the prior works in offline RL.
>
> ## 3. Rejection Sampling
> > Dependence on rejection sampling for policy extraction...
>
> Rejection sampling is beneficial approach that **enables us to avoid backpropagation through time** and has been shown to work on real-world robotics [7]. More generally, offline RL algorithms, by construction, aim to learn the best policy within the coverage of the offline data, so all offline RL algorithms are limited by the coverage of the offline data. However, it should be noted that rejection sampling can exceed the dataset's performance (i.e. the data generating policy's performance).
>
> However, we agree there may be settings where policy gradients are preferable, but **we leave the adaption of our method with policy gradients to future work**. Recent work [9] has integrated policy gradients and flow-based policies by first distilling the expressive, multi-step model into a one-step model. Then, this lightweight one-step model is used to generate actions to optimize the Q function in Q learning. We could leverage a similar approach, where we distill the expressive, multi-step model of our base policy into a one-step model. Then, we could perform Q learning with our learned, optimal Q function in a similar style.
>
> ## 4. Comparisons and Baselines
>
> > Comparative scope is narrow...
>
> We wish to emphasize **the goal of our experimental section: we aim to focus on value learning, specifically investigating whether our method of expressive value learning improves over standard value learning methods**. Comparing to methods like FQL would introduce an orthogonal axis – a separate policy extraction method – that would confound our evaluation of the value learning method. Put simply, **we want our baselines to use the same policy extraction method so that we can target the differences in value learning**.
>
> Moreover, we explicitly state that our aim is not to evaluate the policy extraction methods, and we do not claim that our rejection sampling-based method of policy extraction is the best. We further state that integrating our value learning method into other forms of policy extraction is left for future work.
>
> However, to help address the reviewer’s concerns, **we added IQL [8] as a baseline** because it presents a different method of value learning, specifically one based on expectile regression.
>
> ## Q. D4RL Environment
> > Since the paper emphasizes Scalable Offline Reinforcement Learning, could the authors further validate the effectiveness of Expressive Value Learning on the D4RL MuJoCo datasets...
>
> **We added D4RL tasks [5] to the evaluation**, and **our method achieves the best overall performance on this benchmark** as well.
>
> ## Q. Low-Quality Offline Data
> > Would Expressive Value Learning help improve performance under such more challenging, low-quality offline data distributions...
>
> The **environments we evaluate on [5, 6] offer a mix offline data quality**, including low-quality offline data, and this can be observed by the fact that approaches like behavioral cloning (BC) fail to learn a competitive policy [5]. In contrast, **our method learns a competitive policy that outperforms the baselines**.

---

> > ### Author Response · Authors · 2025-11-25
> >
> > ## Citations
> > [1] Zhou et al. “Q#: Provably Optimal Distributional RL for LLM Post-Training”, 2025
> >
> > [2] Levine et al. “Offline Reinforcement Learning: Tutorial, Review, and Perspectives on Open Problems”, 2020.
> >
> > [3] Morales et al. “A survey on deep learning and deep reinforcement learning in robotics with a tutorial on deep reinforcement learning”, 2021.
> >
> > [4] Ouyang. “Training language models to follow instructions with human feedback”, 2022.
> >
> > [5] Park et al. “OGBench: Benchmarking Offline Goal-Conditioned RL”, 2024.
> >
> > [6] Fu et al. “D4RL: Datasets for Deep Data-Driven Reinforcement Learning”, 2020.
> >
> > [7] Nakamoto et al. “Steering Your Generalists: Improving Robotic Foundation Models via Value Guidance”, 2025.
> >
> > [8] Kostrikov et al. "Offline Reinforcement Learning with Implicit Q-Learning", 2021
> >
> > [9] Park et al. “Flow Q-Learning”, 2025.
> >
> > [10] Farebrother et al., “Temporal Difference Flows”, 2025.

---

### Author Response · Authors · 2025-12-04

We once again thank the reviewers for their time and feedback. We believe we have addressed all of their comments below.

---

### Meta-Review · Area_Chair_GDiT · 2026-01-06

**Summary:**

This paper proposes EVOR, an offline RL framework that shifts “expressivity” from generative policies to generative value learning via flow-matching TD objectives, and performs inference-time policy extraction using rejection sampling with compute scaling (number of samples and temperatures), avoiding BPTT and policy distillation. Reviewers liked the motivation and inference-time scaling behavior, but the decision hinges on concerns about limited novelty/positioning versus closely related distributional/value-learning ideas (e.g., Q#-style log-sum-exp aggregation, TD-Flow/occupancy-flow methods), narrow initial comparisons (mostly QC) and benchmark scope, and the method’s reliance on rejection sampling under suboptimal/sparse datasets, plus questions about deterministic vs stochastic assumptions and missing efficiency reporting.

**Reviewer Concerns:**

The rebuttal addressed several major points by adding D4RL evaluations and an additional strong baseline (IQL), which directly responds to “OGBench-only/QC-only” criticisms, and by clarifying the intent to isolate value-learning improvements under a fixed policy-extraction mechanism. They also clarified novelty as “flow-based TD value learning” (not present in Q#) and argued determinism is a common analysis assumption while EVOR’s TD formulation can be applied more generally; however, one reviewer (PRpU) remained unconvinced because both theory and experiments are still on deterministic tasks and because multi-modality claims for flow-matching policies are not directly validated in the paper. Efficiency comparisons (training time/params) are still not fully reported (authors promise to add runtime later), and some reviewers continue to want stronger comparisons to generative-policy baselines like FQL/diffusion methods and clearer evidence on robustness in low-quality data regimes beyond the added benchmarks.

**Reviewer Scores:**

tSHF likely 4→5 (D4RL + IQL additions address the main request; novelty concerns remain but softened). PRpU likely stays 6→6 (explicitly maintained score due to unresolved stochastic/multi-modality concerns). EQof likely 6→6 (baseline/benchmark additions help; still asks for clearer efficiency and ablations). sTEA likely stays 4→4 (explicitly maintained rating, still wants FQL baseline and runtime/parameter comparisons).
While EVOR is well-motivated and the rebuttal improved the empirical scope (adding D4RL and IQL), the submission still faces persistent concerns on incremental novelty and positioning, incomplete evidence for claimed generality beyond deterministic settings, and insufficient head-to-head comparison/efficiency reporting for a paper whose core claim is “scalable offline RL.” The work is promising, but the current evidence and framing do not yet meet the acceptance bar given the mixed reviewer reception and unresolved key concerns.

---

### Decision · Program_Chairs · 2026-01-26

Reject